# *Paeniclostridium sordellii* hemorrhagic toxin targets TMPRSS2 to induce colonic epithelial lesions

Xingxing Li [1,2,3,4,5], Liuqing He [1,2,3,4,5], Jianhua Luo[2,3,4], Yangling Zheng[2,3,4], Yao Zhou[2,3,4], Danyang Li [2,3,4], Yuanyuan Zhang[2,3,4], Zhenrui Pan[2,3,4], Yanyan Li[2,3] & Liang Tao [1,2,3,4] ✉

Hemorrhagic toxin (TcsH) is an important exotoxin produced by *Paeniclostridium sordellii*, but the exact role of TcsH in the pathogenesis remains unclear, partly due to the lack of knowledge of host receptor(s). Here, we carried out two genome-wide CRISPR/Cas9 screens parallelly with TcsH and identified cell surface fucosylation and TMPRSS2 as host factors contributing to the binding and entry of TcsH. Genetic deletion of either fucosylation biosynthesis enzymes or TMPRSS2 in the cells confers resistance to TcsH intoxication. Interestingly, TMPRSS2 and fucosylated glycans can mediate the binding/entry of TcsH independently, thus serving as redundant receptors. Both TMPRSS2 and fucosylation recognize TcsH through its CROPs domain. By using *Tmprss2⁻/⁻* mice, we show that Tmprss2 is important for TcsH-induced systematic toxicity and colonic epithelial lesions. These findings reveal the importance of TMPRSS2 and surface fucosylation in TcsH actions and further provide insights into host recognition mechanisms for large clostridial toxins.

*Paeniclostridium sordellii* (formerly known as *Clostridium sordellii*) is a gram-positive anaerobic bacterium that opportunistically causes acute infectious diseases in humans and animals, including myonecrosis, enterotoxaemia, sepsis, and toxic shock[1]. *P. sordellii* colonizes the gastrointestinal or vaginal tract of 3–4% of women and is a non-negligible risk factor for obstetric/gynecological procedures[2,3]. For medically reported cases, 74% of the acute *P. sordellii* infections occurred in women undergoing childbirth or abortions with mortality of 100%[4]. The bacterium also frequently infects livestock such as sheep, cattle, and foals, causing severe enteritis, enterotoxaemia, and fatality[1].

Two homologous exotoxins, lethal toxin (TcsL) and hemorrhagic toxin (TcsH), are the dominant virulence factors of *P. sordellii*[5–8]. Both toxins belong to the large clostridial toxin (LCT) family which also includes *Clostridioides difficile* toxin A (TcdA) and toxin B (TcdB), *Clostridium perfringens* large cytotoxin (TpeL), and *Clostridium novyi*

alpha-toxin (Tcnα). TcsH consists of four structural domains: an N-terminal glucosyltransferase domain (GTD) that glucosylates and inactivates Rho-family GTPases, a cysteine protease domain (CPD) that releases the GTD via autocleavage in the cytosol, a combined translocation and receptor-binding domain (DRBD), and a C-terminal combined repetitive oligopeptides (CROPs) domain[8,9]. The CROPs domains of LCTs consist of multiple 19–24 amino acid short repeats (SRs) interspersed with long repeats (LRs) and bear similarity with carbohydrate-binding proteins and may generally interact with various carbohydrate moieties[10–12].

Although TcsH was first reported early in 1969[13], the role of the toxin in *P. sordellii* infections and associated diseases was largely unknown. TcsH is most closely related to *C. difficile* toxin A (TcdA) with a sequence similarity of ~82.8%. TcdA recognizes sulfated glycosaminoglycans and low-density lipoprotein receptor to enter the host cells[14]. Despite being close to TcdA, TcsH does not bind sulfated

[1]Fudan University, Shanghai 200433, China. [2]Key Laboratory of Structural Biology of Zhejiang Province, School of Life Sciences, Westlake University, Hangzhou, Zhejiang 310024, China. [3]Center for Infectious Disease Research, Westlake Laboratory of Life Sciences and Biomedicine, Hangzhou, Zhejiang 310024, China. [4]Institute of Basic Medical Sciences, Westlake Institute for Advanced Study, Hangzhou, Zhejiang 310024, China. [5]These authors contributed equally: Xingxing Li, Liuqing He. ✉e-mail: taoliang@westlake.edu.cn

glycosaminoglycans[15]. Host receptors dictate the cell and tissue specificity for toxin targeting and are critical to dissecting the pathogenesis of the associated diseases. However, no specific cellular receptor(s) has been reported for TcsH, which largely hinders the understanding of its toxin action mechanisms and pathogenesis.

Genome-wide CRISPR/Cas9-based loss-of-function screen is a powerful tool to identify host factors that involve the toxin action. In the pooled library, every cell has a different gene knocked out in principle. The presence of a cytotoxin like TcsH in the pooled library kills the majority of cells, leaving only a minor portion of the survival cells, which normally contains gene knockouts related to the toxin action.

Here, we parallelly performed genetic screens with two different genome-wide CRISPR/Cas9 libraries on HT-29, a human colorectal epithelial adenocarcinoma cell line that is susceptible to TcsH. Both screens revealed TMPRSS2 and fucosylation as critical factors for the cellular entry of TcsH. Because HT-29 cells grow slowly, we later performed the validations in MCF-7, a breast epithelial adenocarcinoma cell line that is also highly sensitive to TcsH. Specifically, we defined that fucosylation mediates robust surface binding of TcsH and TMPRSS2 serves as a specific protein receptor for TcsH. The CROPs domain is essential for TcsH to interact with both fucosylated glycans and TMPRSS2. TMPRSS2 binds to TcsH with a $K_d$ of ~0.13 nM and TcsH CROPs with a $K_d$ of ~5.23 nM. We further investigate the role of TMPRSS2 in the mouse model by using the WT and Tmprss2 knockout C57BL/6 mice. Toxin challenge assay via intraperitoneal injection showed that Tmprss2 knockout mice are more resistant to TcsH compared to WT mice. Finally, using the colon-loop ligation assay, we established TMPRSS2 as a pathologically relevant receptor for TcsH to induce colonic epithelial lesions in vivo.

## Results

### CRISPR screens reveal host factors required for TcsH intoxication

To exam the cell-targeting specificity of TcsH, we set out to assess the toxicity of TcsH in a range of cells and found some cells, including HT-29, Caco-2, and MCF-7, are particularly sensitive to TcsH (Fig. 1a). HT-29, a human colorectal epithelial adenocarcinoma cell line, was chosen for the screen also because the colon is a known target of TcsH. Next, we created HT-29 cells stably expressing spCas9 and further generated CRISPR/Cas9 cell libraries using two independent gRNA libraries (GeCKOv2[16] and TKOv3[17]). TcsH screens were parallelly performed with these two libraries with increasing toxin concentrations for four rounds (Fig. 1b). Four genes, including TMPRSS2, GMDS, FUT4, and SLC35C1, stood out on both screens (Fig. 1c, d, Supplementary Data 1, 2). These genes were not observed in the previous screens for other LCTs and are likely specific for TcsH.

To validate these four genes, we first generated a mixed population of knockout (KO) HT-29 cells using the CRISPR-Cas9 approach to disrupt the gene loci of TMPRSS2, GMDS, FUT4, and SLC35C1. We also picked four less enriched genes (H2AFV, UGT1A9, GPC2, and CNOT1) and generated the mixed HT-29 KO cells. These cells were challenged with TcsH, and the cytopathic effects were measured using the cell-rounding assay. When compared to the parental HT-29 Cas9 cells (referred to as WT), TMPRSS2, GMDS, FUT4, and SLC35C1 KO cells showed drastically increased resistance to TcsH (around 25 to 120-fold), while H2AFV, UGT1A9, GPC2, and CNOT1 KO cells showed no apparent changes in TcsH sensitivity (Fig. 1e, f).

### Cell surface fucosylation mediates the binding of TcsH

Because HT-29 cells grow slowly, we turned to MCF-7, a breast epithelial adenocarcinoma cell line that proliferates faster and is also very sensitive to TcsH (Fig. 1a), and generated single clones of TMPRSS2, GMDS, FUT4, and SLC35C1 KO cells using the CRISPR-Cas9 approach. Consistent with the results obtained in the HT-29 cells, knocking-out

TMPRSS2, GMDS, FUT4, and SLC35C1 also rendered the MCF-7 cells highly resistant to TcsH, as confirmed by both cytopathic cell-rounding assay and immunoblot for RAC1 glucosylation (Fig. 2a and Supplementary Fig. 1).

GMDS, FUT4, and SLC35C1 encode proteins critical for the fucosylation of cell surface glycans. GMDS is a cytosolic enzyme that produces GDP-fucose. SLC35C1 encodes a transporter that specifically imports GDP-fucose into the Golgi apparatus. Inside the Golgi, several fucosyltransferases (FUTs), including FUT4, transfer fucose to assembling glycan structures to generate fucosylated glycans that are later exported to the cell surface[18] (Fig. 2b).

We utilized two fucose-specific lectins, Lotus Tetragonolobus Lectin (LTL) and Aleuria Aurantia Lectin (AAL) to monitor the surface fucosylation of the MCF-7 cells. Little to no fucosylation was detected in the GMDS, FUT4, and SLC35C1 mutants, while the WT and TMPRSS2 mutant exhibited equal levels of glycan fucosylation (Fig. 2c and Supplementary Fig. 2). On the other hand, the MCF-7 WT, GMDS^{-/-}, FUT4^{-/-}, and SLC35C1^{-/-} cells contain similar levels of TMPRSS2, indicating the lack of glycan fucosylation did not affect the expression of TMPRSS2 (Fig. 2d).

We then labeled TcsH with Rhodamine and confirmed that the labeling did not affect its toxicity (Supplementary Fig. 3a). Rhodamine-labeled TcsH strongly binds to the cell surface of the MCF-7 WT and TMPRSS2^{-/-} cells, but not GMDS^{-/-}, FUT4^{-/-}, and SLC35C1^{-/-} cells, suggesting that the fucosylated glycans dominantly mediated the binding of TcsH independent of TMPRSS2 in MCF-7 cells (Fig. 2e).

Because the CROPs from LCTs were reported to have general lectin activity[11,19], we postulated that TcsH recognizes glycan fucosylation also via its CROPs. GFP-fused TcsH CROPs (residues 1832–2618) strongly bound to the MCF-7 WT and TMPRSS2^{-/-} cells, but not the GMDS^{-/-}, FUT4^{-/-}, and SLC35C1^{-/-} cells, which is similar to the full-length TcsH (Fig. 2e). Consistent with the binding result, fucose-specific lectin AAL could competitively protect the MCF-7 cells from TcsH intoxication (Fig. 2f).

To further interrogate the regions within the TcsH CROPs required for fucosylation recognition, we generated several small GFP-fused CROPs fragments (Supplementary Fig. 3b) and tested their binding to the MCF-7 TMPRSS2^{-/-} cells. The C-terminal half of CROPs (residues 2236–2618) strongly binds to the cell surface, while the N-terminal half (residues 1832 to 2246) does not bind (Fig. 2g). However, when TcsH^{2236–2618} is split into three smaller fragments (TcsH^{2229–2413}, TcsH^{2343–2502}, TcsH^{2494–2618}) with each containing one LR, none of them bind to the cells (Fig. 2g). These data suggest that the interaction between TcsH CROPs and surface fucosylated glycans is likely multivalent.

### TMPRSS2 mediates TcsH entry independent of its proteolytic activity

TMPRSS2 encodes a type II transmembrane protein called Transmembrane Serine Protease 2 (TMPRSS2). Ectopic expression of TMPRSS2 in the MCF-7 TMPRSS2^{-/-} cells can well-restore their sensitivity to TcsH (Fig. 3a), confirming that TMPRSS2 contributes to the entry of TcsH. TMPRSS2 is well-known for its ability to facilitate the cellular entry of several viruses, including influenza virus, MERS-COV, SARS-COV, and SARS-COV-2, by proteolytically cleaving the viral envelope glycoproteins[20–23]. To test whether the proteolytic activity of TMPRSS2 involves the TcsH entry, MCF-7, and HT-29 cells were pretreated with Camostat, a serine protease inhibitor that inhibits TMPRSS proteins[24], before the addition of TcsH. The pretreatment with 100 μM Camostat, which was reported to suppress the viral entry[24], did not protect the cells from TcsH (Fig. 3b). We next built a TMPRSS2 mutant (S441A) that has no proteolytic activity[25]. Ectopically expressed TMPRSS2^{S441A} also restored the sensitivity of the MCF-7 TMPRSS2^{-/-} cells to TcsH (Fig. 3a), suggesting that the proteolytic activity of TMPRSS2 is not required for mediating the cellular entry of TcsH.

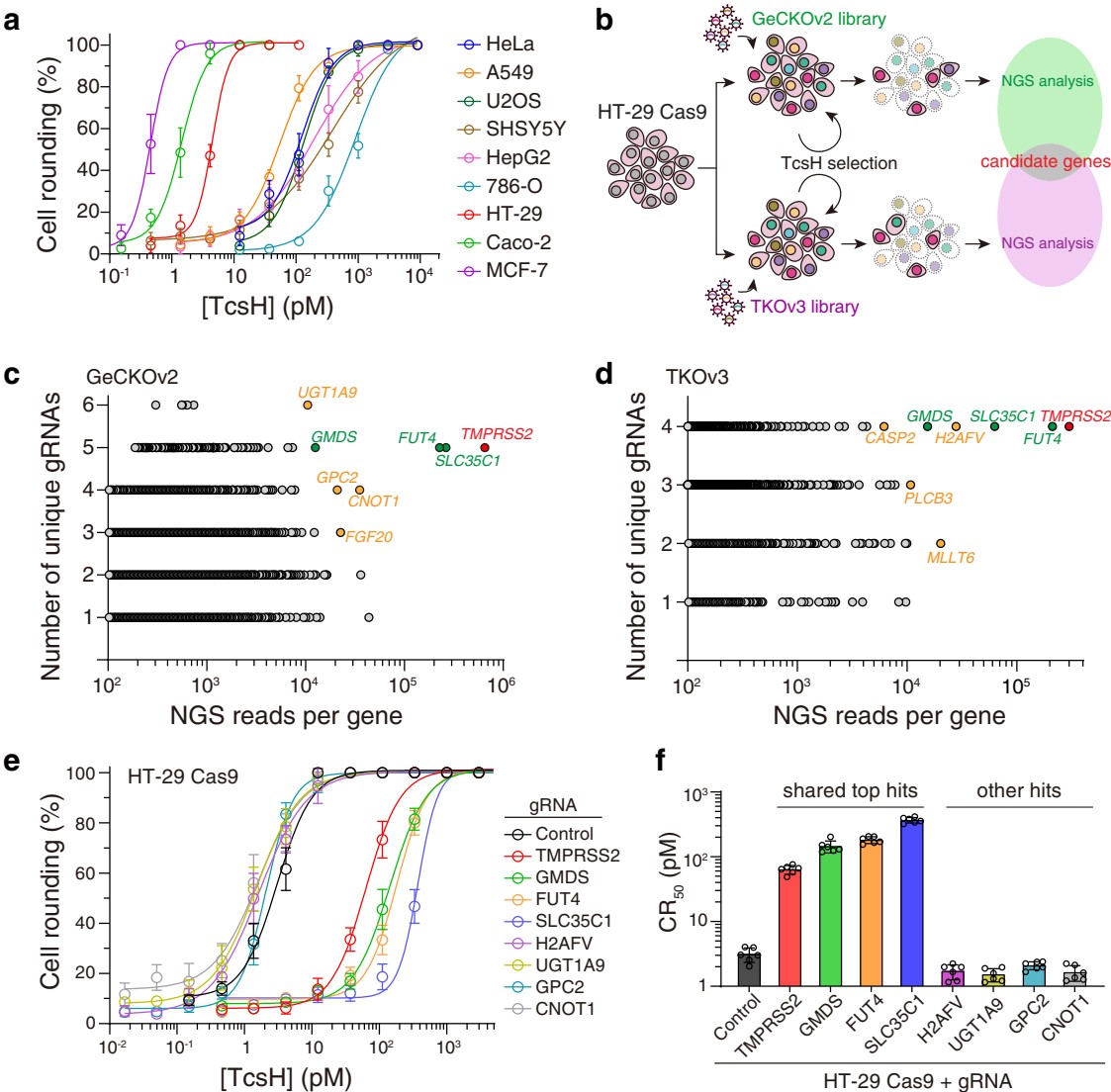

**Fig. 1 | CRISPR screens reveal host factors required for TcsH intoxication. a** The sensitivities of several human cell lines, including HeLa, A549, U2OS, SHSY5Y, HepG2, 786-O, HT-29, Caco-2, and MCF-7, to TcsH were tested using the cytopathic cell-rounding assays. The percentages of the rounded cells were plotted over the TcsH concentrations. **b** Schematic of the screening process using TcsH on HT-29 Cas9 cells. GeCKOv2 and TKOv3 libraries were used independently. Genes enriched after 4 rounds of toxin selection from GeCKOv2 library (**c**) or TKOv3 library (**d**). The *x*-axis represents the number of gRNA reads for each targeted gene. The *y*-axis represents the number of unique gRNA. Top hints were marked with their gene names. **e** The sensitivities of the HT-29 *TMPRSS2*⁻/⁻, *GMDS*⁻/⁻, *FUT4*⁻/⁻, *SLC35C1*⁻/⁻, *H2AFV*⁻/⁻, *UGT1A9*⁻/⁻, *GPC2*⁻/⁻, and *CNOT1*⁻/⁻ cells to TcsH were measured using the cell-rounding assays. The percentages of the rounded cells were plotted over the TcsH concentrations. **f** The measured sensitivities in **e** are quantified and represented as CR₅₀ (the toxin concentration leads to 50% cell rounding after 12–14 h toxin exposure) in a bar chart. In **a**, **e**, **f**, error bars (*n* = 6) indicate mean ± s.d.

TMPRSS2 is expressed in a limited number of cell types according to the public RNA sequencing data ([www.proteinatlas.org](www.proteinatlas.org))[26]. The bioinformatics analysis revealed a correlation between the presence of TMPRSS2 and TcsH susceptibility of the tested cells: MCF-7 and Caco-2 cells are TMPRSS2 positive and sensitive to TcsH; HeLa, U2OS, SHSY5Y, and A549 express little to no TMPRSS2 and these cells are not susceptible to TcsH (Fig. 1a and Supplementary Fig. 4). Ectopically expression of human TMPRSS2, TMPRSS2$^{S441A}$, or mouse Tmprss2 in the HeLa cells largely enhanced their sensitivity to TcsH (Supplementary Fig. 5), supporting the notion that the lack of TMPRSS2 expression is a reason for increased resistance to TcsH in cells like HeLa.

**TcsH selectively interacts with TMPRSS2 with a high affinity**
TMPRSS2 belongs to the serine protease family and is evolutionarily close to other TMPRSS subfamily members[27] (Supplementary Fig. 6). To test the specificity of TMPRSS2 in mediating TcsH entry, we

transiently transfected the HeLa cells with mouse Tmprss2, Tmprss4, Tmprss5, or Tmprss13 and assessed their sensitivity to TcsH. The cytopathic assay showed that only the ectopic expression of Tmprss2, but not Tmprss4, Tmprss5, or Tmprss13, rendered cells increased susceptibility to TcsH (Fig. 3c, d), revealing a high toxin selectivity within the TMPRSS subfamily.

We next managed to detect the direct interaction between TMPRSS2 and TcsH. Because the ectodomain of TMPRSS2 (residues 106–492) can self-cleavage and is not feasible for the exogenous expression and purification, we introduced a mutation (R255Q) into the cleavage site of TMPRSS2 to prevent its autocleavage[25]. Then, we performed a pull-down experiment with Protein A resin and found that Fc-tagged TMPRSS2$^{106-492/R255Q}$ robustly bound to TcsH but not TcsL (Fig. 3e). When supplemented into the cell culture, Fc-tagged TMPRSS2$^{106-492/R255Q}$, but not TMPRSS2$^{106-255}$, effectively blocked the cellular entry of TcsH as demonstrated by the cell-rounding experiment (Fig. 3f and Supplementary Fig. 7a) and immunoblot for RAC1

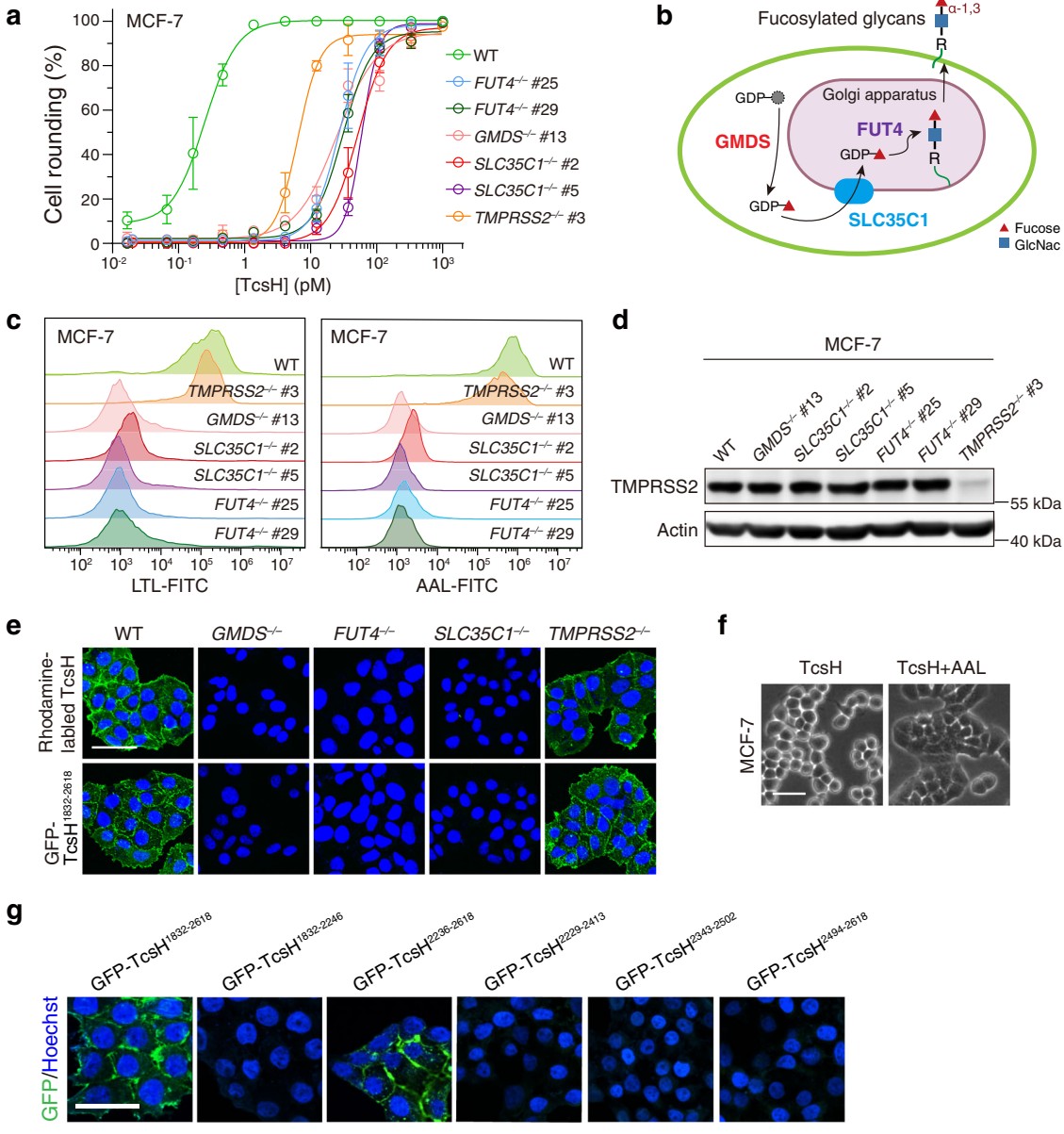

**Fig. 2 | Cell surface fucosylation mediates the binding/entry of TcsH. a** The sensitivities of MCF-7 *TMPRSS2*[−/−], *GMDS*[−/−], *FUT4*[−/−], and *SLC35C1*[−/−] cell lines to TcsH were measured using the cytopathic cell-rounding assay. Error bars represent mean ± s.d., *n* = 6. **b** Schematic view of biosynthesis of fucosylated glycans, genes identified from the screens were highlighted. **c** Flow cytometry profiles of fluorescein isothiocyanate (FITC)-conjugated LTL (left) or AAL (right) binding to MCF-7 WT and KO cells. **d** The absence of TMPRSS2 expression in the MCF-7 *TMPRSS2*[−/−] cells was validated by Western blot analysis. The MCF-7 *GMDS*[−/−], *FUT4*[−/−], and *SLC35C1*[−/−] cells have similar TMPRSS2 expression levels compared to the WT cells. The

experiments have been repeated independently twice with similar results. **e** Confocal fluorescence images show Rhodamine-labeled TcsH (green) or GFP-TcsH[1832−2618] (green) bindings to the MCF-7 WT, *GMDS*[−/−], *FUT4*[−/−], *SLC35C1*[−/−], and *TMPRSS2*[−/−] cells, respectively. Cell nuclei were stained by Hoechst (blue). The scale bar represents 50 μm. **f** Co-incubation of the AAL (8 μg/mL) with TcsH (10 pM, 3.5 h) protected MCF-7 cells from intoxication and prevented cell rounding. **g** Confocal fluorescence images show binding of different GFP-fused TcsH CROPs fragments to the MCF-7 cells. Cell nuclei were stained by Hoechst (blue). The scale bar represents 50 μm.

glucosylation (Fig. 3g). We further confirmed the interaction between TMPRSS2[106-492/R255Q] and TcsH using the biolayer interferometry (BLI) assay, with TMPRSS2[106−255] and TcsL as negative controls (Fig. 3h). The kinetic study revealed a high affinity for TMPRSS2[106-492/R255Q]-TcsH binding with a dissociation constant ($K_d$) of ~0.13 nM (Supplementary Fig. 7b).

**Mapping essential domains for TcsH-TMPRSS2 interaction**

We next managed to investigate the regions involved in the interaction between TMPRSS2 and TcsH. A TcsH truncation lacking the CROPs domain (TcsH[1−1832]) was built and its toxicity on MCF-7 cells was

determined. We showed that TcsH[1−1832] is equally potent to the MCF-7 WT and *TMPRSS2*[−/−] cells (Fig. 4a), indicating the CROPs domain is essential for the recognition of TMPRSS2.

We then tested the binding of Fc-TMPRSS2[106-492/R255Q] to TcsH, TcsH[1−1832], and TcsH[1832-2618] (TcsH CROPs) using the pull-down assay. Fc-TMPRSS2[106-492/R255Q] can readily bind to TcsH but failed to pull-down TcsH[1−1832] (Fig. 4b). Fc-TMPRSS2[106-492/R255Q] could also pull-down TcsH[1832−2618] but the amount seems to be less than TcsH (Fig. 4b). BLI analysis showed that the apparent $K_d$ for TMPRSS2[106-492/R255Q] and TcsH[1832−2618] interaction is approximately 5.23 nM (Supplementary Fig. 7c), which is ~40-fold higher than the $K_d$ value obtained for

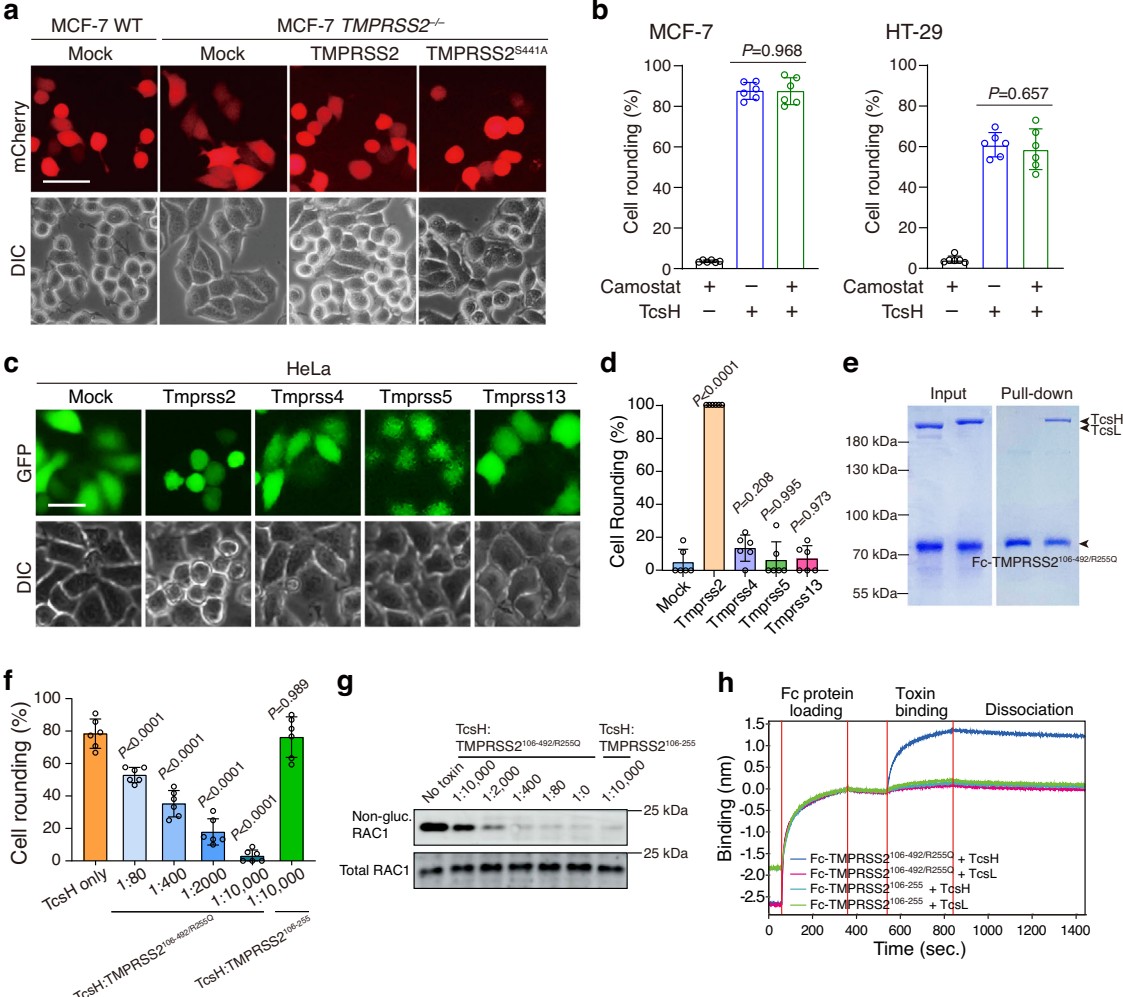

**Fig. 3 | TMPRSS2 is a cellular receptor for TcsH. a** Ectopic expression of either TMPRSS2 or TMPRSS2$^{S441A}$ restored the susceptibility of the MCF-7 *TMPRSS2$^{-/-}$* cells to TcsH. Red fluorescence (mCherry) indicates the transfection. The MCF-7 cells were treated with 10 pM TcsH for 3.5 h. The scale bar represents 50 μm. **b** The MCF-7 or HT-29 cells were pre-incubated with or without 100 μM Camostat (12 h) and then exposed to TcsH (10 pM for MCF-7 and 100 pM for HT-29, 3.5 h). Percentages of rounded cells were quantified and plotted on the bar charts. **c** Ectopic expression of a mouse Tmprss2, but not other Tmprss proteins, enhanced sensitivity of HeLa cell to TcsH (2 nM, 3 h). The scale bar represents 50 μm. **d** The percentages of

round-shaped cells in **c** were quantified and plotted on the bar chart. **e** The pull-down experiment showed that TcsH, but not TcsL, binds to Fc-TMPRSS2$^{106-492/R255Q}$. The Fc-tagged ectodomain of TMPRSS2$^{R255Q}$ protected the MCF-7 cells from TcsH (10 pM, 3.5 h), measured by the cell-rounding assays (**f**) and glucosylation of RAC1 (**g**). The experiments have been repeated independently twice with similar results. **h** Characterization of TcsH and TcsL binding to Fc-tagged TMPRSS2$^{106-492/R255Q}$ or TMPRSS2$^{106-255}$ using the BLI assay. In **b**, **d**, **f**, Error bars represent mean ± s.d., *n* = 6. two-sided Student's *t*-test.

TMPRSS2$^{106-492/R255Q}$ and TcsH. Besides, albeit sequentially similar (similarity of ~73%), the CROPs fragments of TcdA (TcdA$^{1832–2252}$ and TcdA$^{2245–2710}$) had no obvious interaction with TMPRSS2$^{106-492/R255Q}$ as demonstrated by the BLI assay (Fig. 4c).

The ectodomain of TMPRSS2 consists of three distinct structural domains: an LDL receptor class A (LDLRA) domain, a scavenger receptor cysteine-rich (SRCR) domain, and a serine protease domain (SPD)[27]. To interrogate the toxin binding region in TMPRSS2, we generated several TMPRSS2 truncates including TMPRSS2$^{Δ106-149/R255Q}$ (delete LDLRA), TMPRSS2$^{1–254}$ (delete SPD), and TMPRSS2$^{1–149}$ (delete SRCR-SPD) for transient expression in mammalian cells (Supplementary Fig. 8a). TMPRSS2$^{Δ106-149/R255Q}$ can be expressed in the cytosol but failed to reach the plasma membrane (Supplementary Fig. 8b, c). TMPRSS2$^{1–254}$ and TMPRSS2$^{1–149}$ are localized on the cell surface but do not sensitize the HeLa cells to TcsH (Fig. 4d and Supplementary Fig. 8b–d), suggesting the SPD of TMPRSS2 may be necessary for the recognition of TcsH.

## TMPRSS2 and surface fucosylation independently mediate the entry of TcsH

Since we have shown that TcsH-bound fucosylated glycans independent of TMPRSS2 (Fig. 2e), we next managed to address whether TMPRSS2 is capable of mediating the entry of TcsH independent of glycan fucosylation. Ectopic expression of Tmprss2 in the MCF-7 *GMDS$^{-/-}$* and *FUT4$^{-/-}$* cells rendered increased susceptibility to TcsH (Fig. 4e, f). In addition, we further knocked out GMDS from the MCF-7 *TMPRSS2$^{-/-}$* cells and generated the *GMDS$^{-/-}$/TMPRSS2$^{-/-}$* cells. As expected, the MCF-7 *GMDS$^{-/-}$/TMPRSS2$^{-/-}$* cells are more resistant to TcsH than either the *GMDS$^{-/-}$* or *TMPRSS2$^{-/-}$* cells (Fig. 4g), while all these cells are equally sensitive to TcsH$^{1–1832}$ (Fig. 4a). Consistently, residual TcsH binding was observed in the MCF-7 *GMDS$^{-/-}$* cells (images overexposed), and these signals would be further diminished in the *GMDS$^{-/-}$/TMPRSS2$^{-/-}$* cells (Fig. 4h). Together, these results suggest that TMPRSS2 and fucosylated glycans can mediate the binding/entry of TcsH independently.

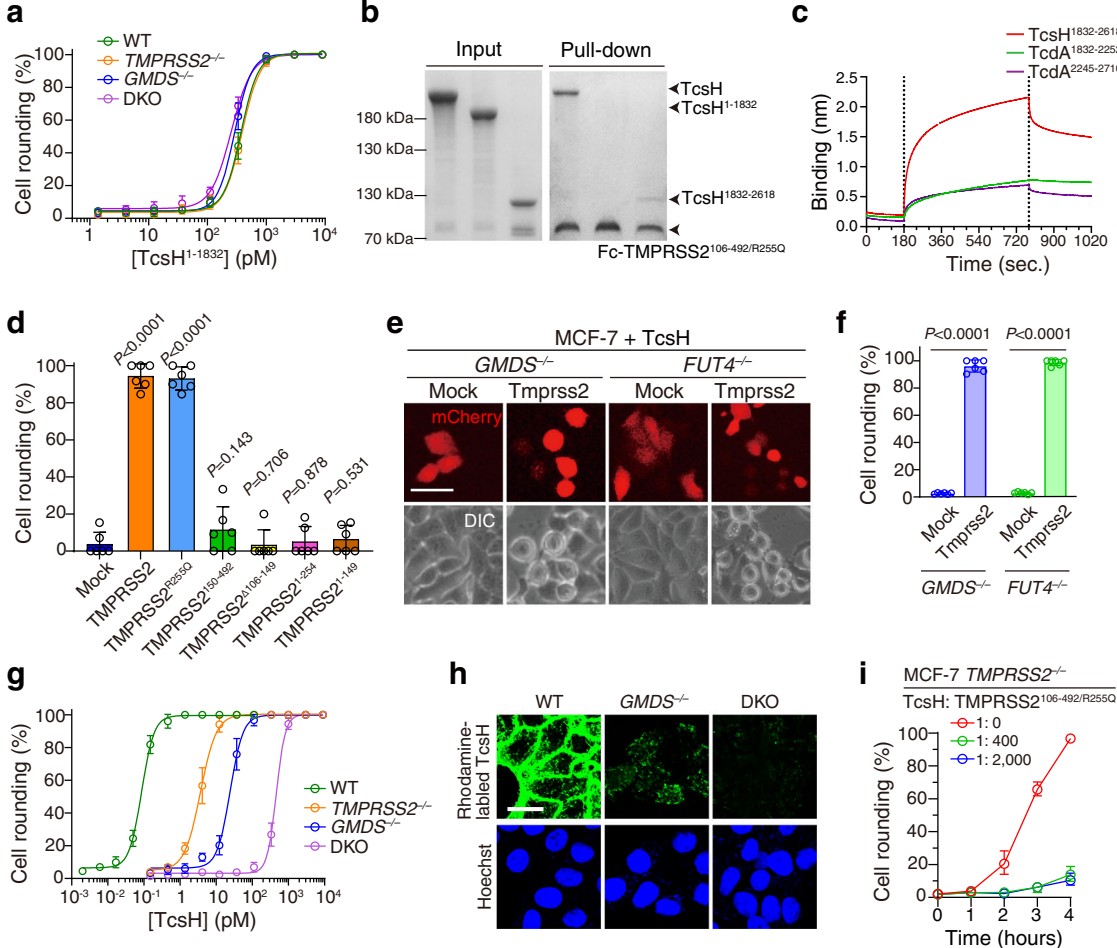

**Fig. 4 | TMPRSS2-TcsH interaction is CROPs-dependent. a** The MCF-7 WT, $GMDS^{-/-}$, $TMPRSS2^{-/-}$, and $GMDS^{-/-}/TMPRSS2^{-/-}$ (DKO) cells showed similar sensitivities to TcsH[1–1832], measured by the cell-rounding assay. **b** The pull-down assay showed that TcsH, but not TcsH[1–1832], binds to Fc-TMPRSS2[106-492/R255Q]. TcsH[1832–2816] is weakly pulled down by Fc-TMPRSS2[106-492/R255Q]. **c** BLI assays showed that TcsH[1832–2618] (1 μM), but not TcdA[1832–2252] (1 μM) or TcdA[2245–2710] (1 μM), binds to Fc-TMPRSS2[106-492/R255Q]. **d** Ectopic expression of TMPRSS2 and TMPRSS2[R255Q], but not others, enhanced the sensitivity of HeLa cells to TcsH. The HeLa cells were treated with 2 nM TcsH for 3 h. The percentages of round-shaped cells were quantified and plotted on the bar chart. Error bars represent mean ± s.d., $n = 6$. two-sided Student's $t$-test. **e** Overexpression of a mouse Tmprss2 further sensitizes the MCF-7 $GMDS^{-/-}$ and $FUT4^{-/-}$ cells to TcsH (10 pM, 10 h). The scale bar represents 50 μm. **f** The percentages of round-shaped cells in **e** were quantified and plotted on the bar chart. **g** The MCF-7 DKO cells are more resistant than either the $GMDS^{-/-}$ or $TMPRSS2^{-/-}$ cells, assayed by cell-rounding count. **h** Confocal fluorescence images (overexposed) show that knocking-out TMPRSS2 in the $GMDS^{-/-}$ cells would diminish the residue TcsH binding. The scale bar represents 20 μm. **i** Fc-TMPRSS2[106-492/R255Q] further protected the $TMPRSS2^{-/-}$ cells from TcsH (100 pM), as showed by the cell-rounding assay over time. The percentage of round-shaped cells was quantified and plotted on the chart. Error bars represent mean ± s.d., $n = 6$.

Notably, TcsH recognizes both TMPRSS2 and surface glycan fucosylation in a CROPs-dependent manner. To determine whether TcsH binds to TMPRSS2 and fucosylated glycans simultaneously or competitively, we used Fc-TMPRSS2[106-492/R255Q] as a competitor and found that it also protected the $TMPRSS2^{-/-}$ cells from TcsH (Fig. 4i), implying that TMPRSS2 and fucosylated glycans are likely competitive receptors for TcsH.

**The CROPs domain is critical for the toxicity of TcsH in vivo**
We then take the advantage of TcsH[1–1832], which does not recognize both TMPRSS2 and fucosylated glycans, to study the potential roles of these receptors in TcsH-induced tissue damage in vivo. When intravenously injected into the mice, TcsH (2 μg/kg) killed C57BL/6 mice in approximately 12 h (Fig. 5a). Mouse laparotomy showed that the liver is an organ being strongly affected and its color turned to dark red (Supplementary Fig. 9a). Hematoxylin and eosin (H&E) stained liver sections revealed massive hemorrhage and necrosis in the hepatic lobules with observable neutrophil infiltrations (Fig. 5b and Supplementary Fig. 9b). In contrast, mice that received TcsH[1–1832] (2 μg/kg) via intravenous injection all survived during the monitoring period (Fig. 5a). Histopathological analysis showed that TcsH[1–1832] did not induce obvious lesions in the liver 8 h post-injection (Fig. 5b and Supplementary Fig. 9b).

**TMPRSS2 is responsible for TcsH-induced damages in the colonic epithelium**
Both fucosylation and TMPRSS2-mediated binding/entry of TcsH is CROPs-dependent and thus could not be dissected using the CROPs-truncated TcsH. To study the physiological role of TMPRSS2 in TcsH-induced tissue damage, we turned to Tmprss2 KO mice, which exhibit no overt developmental defects[28]. When TcsH (2 μg/kg) was injected intravenously, an obvious delay of death (~4 h) was observed in the $Tmprss2^{-/-}$ mice compared to the WT mice with no apparent gender difference (Fig. 5c and Supplementary Fig. 10a). Histopathological analysis showed that TcsH induces severe hepatic hemorrhage and necrosis in both the WT and $Tmprss2^{-/-}$ mice 8 h post injection (Supplementary Fig. 10b).

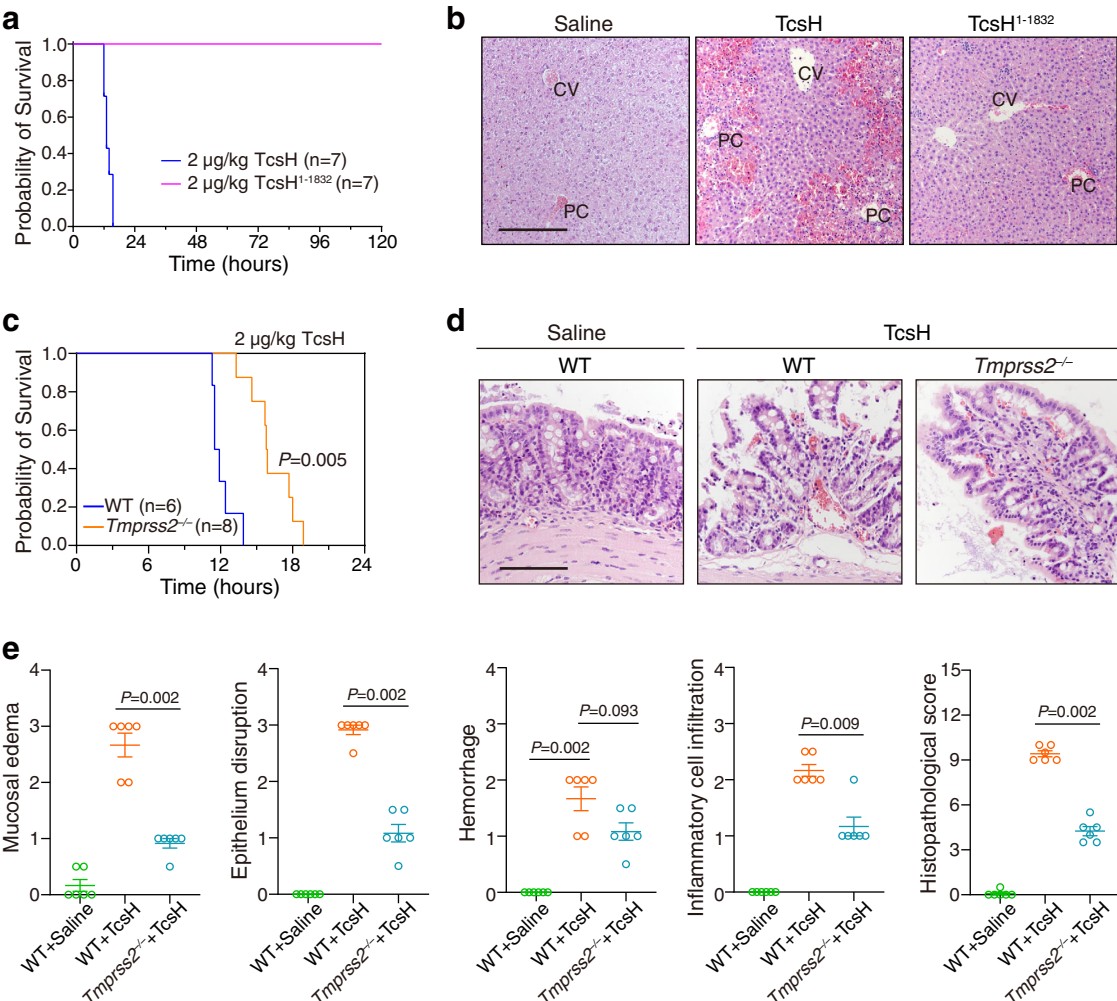

**Fig. 5 | TMPRSS2 is responsible for TcsH-induced colonic lesions.**
**a** Kaplan−Meier curves show the survival of C57BL/6 WT mice intravenously injected with 2 μg/kg TcsH or TcsH[1-1832], respectively. **b** Mouse liver tissues harvested 8 h post tail vein injection of 2 μg/kg TcsH or TcsH[1-1832]. Representative images for the H&E-stained liver sections are from one of three independent experiments (CV: central vein, PC: portal vein). The scale bar represents 200 μm. **c** Kaplan−Meier curves show the survival of C57BL/6 WT or *Tmprss2*[−/−] mice injected with 2 μg/kg TcsH, respectively. Log-rank (Mantel−Cox) test. **d** Mouse colonic tissues harvested after colon-loop ligation assays were assessed for the histopathology induced by 6 μg of TcsH or saline as a control using H&E staining. Representative images are from one of six biological replicates/mice. The scale bar represents 100 μm. **e** Histological scores for **d** were assessed based on mucosal edema, epithelial enterocytes damage, hemorrhage, inflammatory cell infiltration, and overall. *n* = 6 mice per group, error bars indicate mean ± s.e.m., two-tailed Mann−Whitney test.

Extensive studies reported that TMPRSS2 is mainly expressed in certain organs/tissues such as the prostate, kidney, and gastrointestinal tracts, but not in the liver[26,29–31]. We also confirmed that Tmprss2 is expressed in the colonic epithelium but not the liver by the immunohistochemistry (IHC) analysis with the WT and *Tmprss2*[−/−] mice (Supplementary Fig. 10c). Gastrointestinal tracts, particularly the colon and rectum, are common locations where *C. sordellii* colonizes. Therefore, we employed the colon-loop ligation assay to investigate the role of TMPRSS2 in inducing colonic epithelium damage. Histopathological analysis of H&E-stained colon sections showed that TcsH can induce apparent epithelial lesions in the WT mice, as evaluated by mucosal edema, epithelium integration, hemorrhage, and inflammatory cell infiltration (Fig. 5d, e). In comparison, alleviated pathological manifestations caused by TcsH were observed in the *Tmprss2*[−/−] mice (Fig. 5d, e), indicating Tmprss2 is a physiological relevant receptor mediating the TcsH-induced damages in the colonic epithelium.

## Discussion

Recently, the protein receptors of several LCT family proteins, including TpeL, TcdB, TcdA, TcsL, and Tcnα, have been identified[14,15,32–39]. TcsH is the only major member of the LCT family with its host receptor remaining uncharacterized. Here we establish TMPRSS2 and fucosylated glycans as cellular receptors for TcsH and further demonstrate the role of TMPRSS2 in causing systematic toxicity as well as colonic epithelial lesions in vivo. These findings could help to further establish a comprehensive view of how LCTs interact with their host targets and elicit associated diseases.

Cell surface fucosylation is commonly observed in vertebrates, invertebrates, plants, fungi, to bacteria, and could be critical to a broad range of biological processes, including cell adhesion, signaling, and immunity[40,41]. Cell fucosylation was also reported to be involved in some host-pathogen interactions, such as the *Vibrio parahaemolyticus* type III secretion system and H5N2 influenza virus[42,43]. Previous studies demonstrate that the CROPs domain of TcdA is capable of binding to blood group antigens[11,44], which are natural fucosylated carbohydrates linked to lipids or proteins. The carbohydrate-binding sites are shallow troughs consisting of an LR and the following SR based on the structure[19]. Given the high similarity between TcdA and TcsH (~73% for the CROPs domain), it would not be surprising that the CROPs of TcsH can readily bind to cell surface fucosylated glycans. Notably, TcsH

seems to be simultaneously bound by multivalent fucosylated glycans. This would allow the toxin to achieve a high-affinity attachment to its target cell surface, which is a common strategy for many bacterial toxins such as Shiga toxin[45].

TMPRSS2 was originally identified because its expression can be strongly induced by androgens in prostate cancer[29]. This membrane-anchored serine protease is also well-known for its ability to cut the glycoproteins of several famous respiratory viruses and activate the internalization[20–23]. Nevertheless, we found that its proteolytic activity is dispensable for mediating the cellular entry of TcsH and the TMPRSS2 can directly bind to TcsH with low nanomolar affinity. Thus, the mechanisms of TMPRSS2-mediated entry for TcsH and viruses are distinct.

By using the TcsH fragments, we showed that TMPRSS2 binds to the CROPs domain of TcsH. But the detected TMPRSS2-TcsH$^{1832–2618}$ binding ($K_d$: ~5.23 nM) is lower than the TMPRSS2-TcsH binding ($K_d$: ~0.13 nM), possibly other regions of the toxin also contribute to the interaction. On the other side, we found that the SPD is indispensable for TMPRSS2 to function as a cellular receptor for TcsH.

Our results clearly showed that TMPRSS2 and surface fucosylation can independently mediate the binding/entry of TcsH and they may serve as competitive receptors. Fucosylation-mediated TcsH binding is dominant in MCF-7 cells, which is likely due to the abundance of surface glycan fucosylation. In addition, our data demonstrated that the *TMPRSS2$^{–/–}$/GMDS$^{–/–}$* cells are more resistant to TcsH compared to the *TMPRSS2$^{–/–}$* or *GMDS$^{–/–}$* cells, suggesting TMPRSS2 and fucosylated glycans can functionally serve as redundant receptors for TcsH. Since fucosylated glycans and TMPRSS2 have varied distribution and abundance in different cell types[30], the adoption of redundant receptors may allow TcsH to access a broad range of host targets.

TMPRSS2 is responsible for the TcsH-induced colonic epithelial lesions, as most of the pathological features, including mucosal edema, epithelium integration, and inflammatory cell infiltration, are reduced in the *Tmprss2$^{–/–}$* mice compared to the WT mice. Because intestinal lumens are major locations where *P. sordellii* colonizes in humans[2,46], TcsH actions may benefit the bacterium by damaging the colonic epithelium to gain additional nutrients. Indeed, HT-29 and Caco-2, both colorectal epithelial carcinoma cell lines tested, are highly sensitive to TcsH. However, the full-length TcsH is still potent in the *Tmprss2$^{–/–}$* mice and can elicit severe hemorrhage in the liver, which is different from the phenotypes observed in TcsH$^{1–1832}$ injected WT mice. Therefore, we propose that fucosylation is also critical for mediating TcsH entry in vivo or potentially other CROPs-dependent or -independent receptors remain.

The CROPs domain is critical for the toxicity of TcsH in both cultured cells and mouse models, which somehow differs from other LCTs[14,35,47]. Particularly, we showed that the CROPs domain of TcsH alone can interact with TMPRSS2 with nanomolar affinity, while all other reported protein receptors for LCTs bind to regions beyond the CROPs (TcdB-CSPG4 binding is CROPs-dependent but only a small junction region between the DRBD and CROPs is involved)[14,36,48–50]. The findings that TMPRSS2 binds to the CROPs of TcsH could provide a new model for LCT to recognize the host protein receptors. It may also be worth reassessing the potential of the CROPs from other LCTs to bind host protein receptors besides sugar moieties.

## Methods

### Materials

All cell lines were originally obtained from ATCC except for Expi293F, which is purchased from ThermoFisher Scientific (U.S.). HeLa (H1, CRL-1958), HT-29 (HTB-38), and HEK293T (CRL-3216) cells were authenticated via STR profiling (Shanghai Biowing Biotechnology Co. Ltd, Shanghai, China). Expi293F cells were cultured in SMM 293-T II Expression Medium (Sino Biological, Beijing, China). All other cells were cultured in DMEM supplemented with 10% fetal bovine serum

(FBS) and 0.1 mg/mL streptomycin/ penicillin, in a humidified atmosphere of 95% air and 5% $CO_2$ at 37 °C. The following antibodies and reagents were purchased from the commercial vendors: mouse monoclonal antibody against non-glucosylated RAC1 (Clone 102, #610650, BD Biosciences, 1:1000), total RAC1 (Clone 23A8, MA1-20580, Invitrogen, 1:1000), and Flag-tag (Clone 5A8E5, A01809, GenScript, 1:200), rabbit monoclonal antibody against TMPRSS2 (Clone EPR3862, ab109131, Abcam, 1:2000 for Western blot) and β-actin (Clone AC-15, #078M4809V, Sigma, 1:5000), rabbit polyclonal antibody against TMPRSS2 (14437-1-AP, Proteintech, 1:1000 for IHC), horseradish peroxidase-labeled goat anti-rabbit IgG (H + L, PI-1000, Vector Labs, 1:10000 for Western blot), horseradish peroxidase-labeled goat anti-mouse IgG (H + L, PI-1000, Vector Labs, 1:10000 for Western blot), Precast PAGE Gel (abs9308, Absin), Hoechst 33258 (E607301, BBI), LTL-FITC (FL-1321, Vector Laboratories), AAL-FITC (FL-1391, Vector Laboratories), Camostat mesylate (HY-13512, MCE), Poly-ethylenimine Linear (40816ES03, Yeasen), and NHS-Rhodamine fluorescent labeling kit (#46406, ThermoFisher Scientific).

### Mice

C57BL/6 mice were purchased from Laboratory Animal Resources Center at Westlake University (Hangzhou, China). Tmprss2 KO mice were purchased from GemPharmatech (Nanjing, China). Male and female, 6–8 weeks C57BL/6 WT and Tmprss2 KO mice were used in this study. Mice were housed in specific-pathogen-free micro-isolator cages with free access to drinking water and food and monitored under the care of full-time staff. All mice had a 12-h cycle of light/darkness (7 a.m. to 7 p.m.), housed at 20–24 °C with 40–60% humidity.

### cDNA constructs

The sgRNA sequences targeting *TMPRSS2*, *GMDS*, *SLC35C1*, *FUT4*, *CNOT1*, *H2AFV*, *GPC2*, and *UGT1A9* were cloned into the LentiGuide-Puro vector. The cDNAs of human TMPRSS2 protein and mouse Tmprss2 family proteins were PCR amplified and cloned into the pLVX-mcherry vector. The *TcsH* gene was codon-optimized and synthesized by a commercial vendor (Genscript, Nanjing, China). Gene fragments encoding TcsH and TcsH$^{1–1832}$ were PCR amplified and cloned into the pHT01 vector. Gene fragment encoding TcsH$^{1832–2618}$, TcsH$^{1832–2246}$, TcsH$^{2236–2618}$, TcsH$^{2229–2413}$, TcsH$^{2343–2502}$, and TcsH$^{2494–2618}$ were fused with GFP at the N-termini and cloned into the pET28a vector.

### Recombinant proteins

Recombinant TcsH and TcsH$^{1–1832}$ were expressed in *Bacillus Subtilis* and purified as His-tagged proteins as previously described. GFP-tagged TcsH$^{1832–2618}$, TcsH$^{1832–2246}$, TcsH$^{2236–2618}$, TcsH$^{2229–2413}$, TcsH$^{2343–2502}$, and TcsH$^{2494–2618}$ were expressed in *Escherichia coli* and purified as a His-tagged protein. The recombinant human Fc-tagged chimera proteins with His tag at N-terminal were cloned into a pHLsec vector and expressed using Expi293F cells: Fc-TMPRSS2$^{106-255/R255Q}$ and Fc-TMPRSS2$^{106-492/R255Q}$. Briefly, $5 \times 10^8$ Expi293F cells were transfected with 750 μg plasmid using Polyethylenimine Linear (1 mg/mL). The culture was harvested 4 days after transfection. The proteins in the culture medium were collected and purified as His-tagged proteins.

### Genome-wide CRISPR-Cas9 screens

Two CRISPR/Cas9 genome-wide KO lentiviral libraries were generated based on the GeCKOv2 library or TKOv3 library respectively. GeCKOv2 is composed of 123,411 gRNAs in two sub-libraries. Each sub-library contains three unique sgRNAs per gene. The TKOv3 contains 70,948 gRNAs targeting 18,053 protein-coding genes (four gRNAs per gene). HT-29 Cas9 cells were transduced with sgRNA lentiviral library at a multiplicity of infection of 0.2. For each cell library, $7.9 \times 10^7$ cells were plated onto three 15-cm cell culture dishes to ensure sufficient sgRNA coverage, with each sgRNA being represented around 1200 times. These cells were exposed to TcsH for 12–18 h and then washed three

times to remove loosely attached cells. The remaining cells were cultured with a toxin-free medium to ~70% confluence and subjected to the next round of screening with higher concentrations of toxins. Four rounds of screenings were performed with TcsH (20, 50, 100, and 200 pM). The remaining cells from each round were collected and their genomic DNA was extracted using the Blood and Cell Culture DNA mini kit (Qiagen). DNA fragments containing the sgRNA sequences were amplified by PCR using primers lentiGP-1_F (AATGGACTATCA TATGCTTACCGTAACTTGAAAGTATTTCG) and lentiGP-1_R (TAAAAAA GCACCGACTCGGTGCCACTTTTTCAAG). The next-generation sequencing was performed by a commercial vendor (Novogene, Beijing, China).

### The cytopathic cell-rounding assay

The cytopathic effect of the toxin was analyzed using the gold-standard cell-rounding assay. Briefly, cells were exposed to toxins for 12–14 h. The phase-contrast images of the cells were captured by a microscope (Olympus IX73; ×10 or ×20 objectives) with the software Olympus CellSens Standard 2.1. Six zones of 300 μm × 300 μm were selected randomly, with each zone containing ~50–250 cells. Round-shaped and normal-shaped cells were counted manually. The percentage of round-shaped cells was analyzed using GraphPad Prism (ver. 9.0.0, GraphPad Software, LLC).

### Generating KO cells

LentiCas9-Blast (Addgene) was used to generate lentiviruses and subsequentially transduced into HT-29 or MCF-7 cells to produce HT-29 Cas9 or MCF-7 Cas9 cells. To generate the TMPRSS2, GMDS, SLC35C1, FUT4, GPC2, CNOT1, H2AFV, and UGT1A9 KO cells, the following sgRNA sequences were cloned into the LentiGuide-Puro vector (Addgene) to target the indicated genes: 5′-ACTGTGCATCACCTT GACCC-3′ (TMPRSS2), 5′-GATGGGCAAGCCCAGGAACG-3′ (GMDS), 5′-AACCTCTGCCTCAAGTACGT-3′ (SLC35C1), 5′-TCTATCGCCGCTACTT CCAC-3′ (FUT4), 5′-AAGCAGGAGAGGTCGCAGCG-3′ (GPC2), 5′-TCTTG GTTAAATTGTCCACC-3′ (CNOT1), 5′-AAGACTCGCACCACAAGCCA-3′ (H2AFV) and 5′-TGGAGTGACCCTGAATGTTC-3′ (UGT1A9). HT-29 Cas9 or MCF-7 Cas9 cells were transduced with lentiviruses expressing the sgRNAs and selected with puromycin (1 μg/mL for HT-29 and 2.5 μg/mL for MCF-7). For MCF-7 KO cells, single colonies were further isolated and validated.

### Biolayer interferometry (BLI) assay

The binding affinities between toxin fragments and TMPRSS2 proteins were measured using the BLI assay with the Octet RED96e system and analyzed with the Octet Data Analysis software (version 12.0.1.2, ForteBio, Fremont, CA, USA). Briefly, Fc-tagged proteins were immobilized onto the capture biosensors (AHC biosensor, ForteBio) and balanced with binding buffer (20 mM Tris-Cl, 150 mM NaCl, pH = 7.4). The biosensors were then exposed to the indicated concentrations of TcsH, TcsL, TcsH$^{1-1832}$, TcsH$^{1832-2618}$, TcdA$^{1832-2252}$, or TcdA$^{2245-2710}$, followed by dissociation in binding buffer.

### Cell surface binding assays

MCF-7 cells were incubated with 10 μg/mL FITC-AAL, 10 μg/mL FITC-LTL, 50 nM Rhodamine-labeled TcsH, or 50 nM GFP-TcsH$^{1832-2618}$ in the medium on ice or at room temperature for 20 min. Cells were washed twice with ice-cold PBS, fixed with 4% paraformaldehyde (PFA) for 15 min at room temperature, stained with or without Hoechst, followed by fluorescence microscopy or flow cytometry analysis. Fluorescent images were captured using an Olympus FV3000 inverted LSCM Confocal System with the software FV31S-SW v2.3.2.169. For flow cytometry, cells were trypsinized, washed with PBS, resuspended in carbo-free blocking solution, and incubated for 30 min at room temperature. The cells were then washed and stained with 10 μg/mL FITC-LTL or 10 μg/mL FITC-AAL for 20 min at 4 °C, washed twice with cold

blocking solution, and resuspended in PBS. Flow cytometry was carried out on a CytoFLEX LX (Beckman Coulter) and the data were analyzed with the software CytoExpert v2.4 (Beckman Coulter) and FlowJo v.10.8.0 (BD Biosciences).

### Cell surface immunofluorescent staining

MCF-7 cells were fixed with 4% PFA for 15 min at room temperature and then washed twice with PBS. Cells were blocked with 1% BSA in PBS for 30 min at room temperature but were not permeabilized. Cells were then incubated with Flag-tag antibody (1:200, conjugated with Alexa Fluor 488) for 1 h at room temperature. Cells were washed with PBS three times, followed by staining with Hoechst. Fluorescent images were captured using an Olympus FV3000 inverted LSCM Confocal System with the software FV31S-SW v2.3.2.169.

### Competition assay with TMPRSS2 proteins

TcsH were pre-mixed with or without recombinant Fc-tagged TMPRSS2 proteins in the fresh culture medium and incubated on ice for 30 min. The mixtures were then added to the cell culture. Cells were further incubated at 37 °C and the percentages of rounded cells were recorded.

### Toxin challenge assays in mice

Six- to eight-week-old male and female mice were intravenously injected with 2 μg/kg TcsH or 2 μg/kg TcsH$^{1-1832}$ respectively. The animals were monitored for up to 5 days post challenge for toxic effects and mortality, and mice were killed if they became moribund. Survival was graphed as Kaplan–Meier curves. For histopathological studies, the mice were injected with the toxins and then euthanized after 8 h. The livers were excised out, fixed, paraffin-embedded, sectioned, and subjected to H&E staining.

### Colon-loop ligation assay

Six- to eight-weeks-old male mice were anesthetized by intraperitoneal injection of 1% pentobarbital sodium. A midline-right laparotomy was performed to locate the ascending colon and seal a ~2 cm loop with 4-0 surgical suture ligatures. Six micrograms of TcsH in 100 μL of normal saline or 100 μL of saline alone was injected into the sealed colon segment using an insulin syringe, followed by suturing of the skin incision. Mice were allowed to recover in the 37 °C thermostatic plates. After 8 h, mice were euthanized, and the ligated colon segments were excised. The colon segments were fixed, paraffin-embedded, sectioned, and subjected to either H&E staining for histological scoring.

### IHC, H&E staining, and histopathological analysis

Colon and liver specimens were fixed in a 4% formaldehyde aqueous solution for 12 h before dehydration with gradient alcohol. The samples were then cleared with xylene and embedded in paraffin. Paraffin blocks were cut into 5 μm sections and stained by H&E or IHC. For IHC, antigen retrieval was performed by boiling the sections in the retrieval buffer (E-IR-R104, Elabscience) for 15 min. The sections were treated with 3% $H_2O_2$, blocked with 10% sheep serum, and incubated overnight with a primary antibody against Tmprss2 (1:1000). The slides were then washed with PBS three times. Immunoreaction was performed with a Two-Step Plus Poly-HRP Anti-Rabbit/Mouse IgG Detection Kit (E-IR-R213, Elabscience) following the manufacturer's instruction. Cell nuclei were stained with hematoxylin. The H&E staining sections were scored blinded by two pathologists based on edema, inflammatory cell infiltration, epithelium disruption, and hemorrhage on a scale of 0 to 3 (mild to severe). The average scores were plotted on the charts.

### Ethics statement

All animal procedures reported herein were performed following the institutional guidelines and approved by the Institutional Animal Care and Use Committee at Westlake University (IACUC Protocol

#19-010-TL). To minimize the distress and pain, the mice injected with toxins were monitored every hour. Animals with signs of pain or distress such as labored breathing, inability to move after gentle stimulation, or disorientation were euthanized immediately. This method was approved by the IACUC and monitored by a qualified veterinarian.

## Reporting summary

Further information on research design is available in the Nature Research Reporting Summary linked to this article.

## Data availability

The data that support the findings of this study are available from the corresponding author upon reasonable request. Source data are provided with this paper.

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

## Acknowledgements

We thank Dr. Yongdeng Zhang for the assistance in microscopy and members of the Tao laboratory for the discussions. We are also grateful to the Biomedical Research Core Facilities and Laboratory Animal Resources Center of Westlake University for providing technical support. This study was partially supported by the National Natural Science Foundation of China (Grant no. 31970129 to L.T. and Grant no. 32171205 to Y.L.) and the Zhejiang Provincial Natural Science Foundation of China (Grant no. LR20C010001 to L.T. and Grant no. LR20C050001 to Y.L.). L.T. also acknowledges support from Westlake Education Foundation and Westlake Laboratory of Life Sciences and Biomedicine.

## Author contributions

L.T. conceived the project and designed the experiments. X.L. and L.H. performed CRISPR screens, cell biology and biochemical experiments, and data analysis. X.L., L.H., D.L., and Y.L. purified the proteins. J.L., X.L., and Y. Zhang performed animal experiments and histological analysis. Y. Zheng, Y. Zhou, and Z.P. helped with the plasmid construction and cell-based experiments. L.T. wrote the manuscript with input from all co-authors.

## Competing interests

The authors declare no competing interests.

## Additional information

**Peer review information** *Nature Communications* thanks other anonymous reviewer(s) to the peer review of this work. Peer review reports are available.

