## [Peer Review File · Nature Communications]

REVIEWER COMMENTS

Reviewer #1 (Remarks to the Author):

The *Paenibacillus sordellii* toxin TcsH belongs to the family of clostridial glucosylating toxins and, in contrast to the other members of this toxin family, receptors for this toxin have not been described yet. Li and colleagues were able to identify TMPRSS2 as a receptor for TcsH with an elegant CRISPR/Cas9-based approach. In addition, they could show that cell surface fucosylation contributes to binding and entry of TcsH into cells.

The main findings of this study are convincing and of high relevance in the field of bacterial toxins. By and large, the experiments are well-performed with appropriate controls. It is also of note that the manuscript is well-written and the figures of high quality, adhering to the high standards of the journal.

However, several issues need to be addressed, before considering acceptance of this sophisticated manuscript for publication.

Main text:

- Line 51 and 58: Two abbreviations (LCGT, LCT) are used for the same toxin family.
- Line 95 and 98: It is not entirely clear to me, what is meant with "mixed population".
- Line 129: "Extended Data Fig. 3" indicated here are not explained in the main text.
- Line 131: "...fucosylation-mediated TcsH binding does not require TMPRSS2"; is the opposite also the case?
- Line 147: "...susceptibility of the tested cells to TcsH"; such data about the susceptibility mentioned here are missing in the manuscript; what is known about fucosylated glycans in these cell lines mentioned here?
- Line 151: Unclear, why the term Tmprss2 is in small letters here, but elsewhere in capital letters.

- Line 157: “into the HT-29 cells”; why did the authors test Camostat (only) in this cell line? It is a bit confusing that in most experiments MCF-7 cells are used, but in this case HT-29 cells. Also HeLa cells are used in this manuscript, without explanation from the authors why they switched to these cells.

- Line 158: “...Camostat did not show any protection of the cells...”; positive control for the inhibitory action of Camostat is missing. It must be shown that Camostat is functional in the concentration used here. I am not convinced from the current data.

- Lines 179-186: The finding that exclusively TMPRSS2 and no other family members serve as receptor for TcsH is not very surprising, since the CRISPR/Cas9 screening method is based on single-gene knockouts. If other family members would act as receptors, isolation of a TMPRSS2 KO mutant would not be possible. This should be mentioned here or in the Discussion section.

In addition, it is unclear to me, why the authors did not perform this reconstitution experiment with the CRISPR/Cas9-derived, defined TMPRSS2 KO cell line? Such a reconstitution experiment is of major importance in order to exclude off-target effects in CRISPR/Cas9-derived TMPRSS2 KO cells.

- Line 192: “...TcsH and TcsH1-1832 are equally potent to these cells...”; TcsH is ~1000-fold more potent in MCF-7 cells (Figure 2a), isn't it?

TcsH1-1832 is equally potent in MCF-7 GDMS^{-/-} and TMPRSS2^{-/-} cells. This indicates an additional CROP-independent receptor, which should be discussed.

- Line 194: What are the two TcdA constructs shown in Figure 4b (green and purple)? Not all constructs explained in the text are shown in the Figure 4b.

- Lines 196-197: “...interaction between TMPRSS2 and TcsH1-1832 197 was not detected.”; by which method? No data are shown. The pulldown approach shown in Figure 3e could be used to confirm the claim.

- Line 197: “...sequentially similar..”; please provide the percentage in similarity.

- Line 204: “...all failed to facilitate the TcsH entry...”; did the authors check, if the truncated proteins are able to reach the plasma membrane and if they are expressed in similar levels? Without this information, it is not possible to know, if the deleted parts in the truncated version are relevant for the interaction with TcsH.

- Line 242-243: "...TcsH also induces severe hepatic hemorrhage and necrosis in the *Tmprss2*^{-/-} mice."; which Figure includes this data? Images of the liver are not shown like in Extended Figure 9a for wt mice.

- Lines 244-246: "TMPRSS2 is mainly expressed in certain organs/tissues such as the prostate, kidney, and gastrointestinal tracts, but not the liver."; but why is then the liver harmed by TcsH as shown in Extended Figure 9a? Can the authors confirm the lack of TMPRSS2 expression in the mouse liver by immunoblotting?

- Line 252-253: "...alleviated pathological manifestations caused by TcsH were observed in the *Tmprss2*^{-/-} mice."; since the authors used commercially available *Tmprss2*^{-/-} mice, it still would be desirable to confirm lack of *Tmprss2* expression in the colonic epithelium by immunoblotting.

Figures:

- Figure 2c: Can the lectins used here compete with toxin binding and inhibit TcsH intoxication?

- Figure 2d: Why is there still a faint band for TMPRSS2 in the last lane for the TMPRSS2-KO mutant? This indicates that the authors might use CRISPR/Cas9-generated KO mutants that are not homozygous. T7E1 assays should be performed to clarify if the mutants are homozygous with biallelic KO of TMPRSS2.

- Figure 2e: Why does TcsH not bind to cells lacking glycan fucosylation although TMPRSS2 is expressed in those cells (Figure 2d)? Binding experiments with the constructs shown here should be repeated with the double KO (DKO) mutant shown in Figure 4f.

- Figure 2f: Why did the authors not intoxicate TMPRSS2 knockout cells with TcsH1-1832? This would be of interest.

- Figure 3a: This experiment should be repeated with the MCF-7 TMPRSS2-KO cells, since HeLa cells do still express TMPRSS2 and cleavage of TMPRSS2(S441A) could occur by intermolecular cleavage. In addition, 2 nM TcsH were used in HeLa cells, a very high concentration, to achieve cell rounding already after 2 hours. At such conditions and without time kinetics, it is difficult to see any difference in the sensitivity between TMPRSS2- and TMPRSS2(S441A)-expressing HeLa cells against TcsH.

- Figure 2c: The negative control TcsH:TMPRSS2(106-255) is not mentioned or described in the main text.

- Figure 3e. It would be great to see TcsH CROPS and TcsH1-1832 in this pull-down approach, since both construct are available.

- Figure 3f: The fitting curve should be shown that was used to estimate the dissociation constant.

- Figure 3f: Please use "sec" (Fig. 3f) or "s" (Fig. 4b) in the x-axis uniformly.

- Figure 4a: A citation in the text is missing.

- Figure 4b: Why is the binding of TcsH1833-2618 here twice as high when compared to Extended Figure 7c?

- Figure 4f: It would be great to test also TcsH1-1832 and TcdA here. Especially for TcsH1-1832, it would be interesting to know, if it is equally potent in GMDS KO cells and DKO cells when compared to wildtype cells as shown for the TMPRSS2 KO cells in Figure 4a.

- Extended Figure 7b and 7c: For uniformity reasons, please include here also the amino acid numbers of the ectodomains (after TMPRSS2) as provided in Figure 3f.

Reviewer #2 (Remarks to the Author):

This manuscript describes experiments aimed at identifying genes that encode proteins and or glycans on human cells that could act as receptors for haemorrhagic toxin (TcsH), one of the two toxins produced by the pathogenic bacterium *Paenibacillus sordellii* (previously known as *Clostridium sordellii*). *P. sordellii* is an important pathogenic bacterium known to cause a variety of soft tissue infections, notably in the gastrointestinal tract of animals and in the vaginal tract of humans. While

receptors have been identified for related toxins produced by *P. sordelli* (TcsL) and *Clostridioides difficile* (*C. difficile*), to my knowledge no receptors have been identified for TcsH.

Introduction. There is an assumption that all readers will know what a CRISPR screen is and the rationale for the experiments conducted. This is not necessarily the case. The authors should state that the purpose of the screens is to select for mutants that are resistant to the effects of the toxin, and that some of these mutants will be deficient in expression of a surface component that recognises TcsH toxin. It would also be helpful to mention that HT29 cells are a human colorectal adenocarcinoma cell line. Last paragraph on Introduction is poorly written. It is essentially a repeat of the essential conclusions reached in the manuscript. In addition, there is no mention in the Introduction of the species used as the experimental model.

Fig 1. The approach used by the authors was to create libraries of mutants using CRISPR/Cas9 in the toxin-sensitive HT-29 cell line and to select for mutants resistant to TcsH. Two independent gDNA libraries were used to create two cell libraries. The screening experiments revealed four promising hits and individual HT-29 knock-out mutants showed increased resistance to TcsH, suggesting these gene products were involved in cellular intoxication.

Fig 2. Similar results were obtained when a different cell line, MCF-7, was used to create k/o mutants. Three of the identified genes (GDMS, FUT4 and SLC35C1) encode proteins involved in either the biosynthesis of fucose (GDMS), its transport (SLC35C1) or its transfer to glycans i.e. a glycosyltransferase). Two fucose-specific lectins were used to demonstrate that in the GDMS, FUT4 and SLC35C1 mutants, considerably lower levels of fucose were detected, but normal levels were seen in the TMPRSS2 mutant. Little or no binding of Rhodamine-labelled TcsH was seen in the fucose-deficient mutants, but high levels (but not as high as wild type) of binding was seen in the TMPRSS2 mutant. Using toxin-deletions, the authors showed that the fucose binding was mediated by the CROP domain, a result consistent with previous data showing lectin-like activity of the related CROP domains of TcdA toxin from *C. difficile*. Finally a CROP-deletion mutant toxin had approx. 3 log reduced toxicity against WT cells, a result also seen with the GDMS mutant. Together these results suggest that either a fucose-containing glycan or glycoprotein acts as a receptor for TcsH toxin.

Fig 3. The authors then go on to investigate the fourth gene they identified, encoding the transmembrane serine protease MPRSS2. In Fig 3. They use HeLa cells that do not express MPRSS2 transfected with MPRSS2. Fig 3A shows some cell rounding but is somewhat redundant as it is not quantitative. A modest 60% cell rounding at a high (2 nM) toxin level. Using either Camostat (a serine protease inhibitor) or a catalytic mutant of MPRSS2 in these experiments showed that the serine protease activity was not required for receptor activity. Entry of toxin into cells was reduced on ectopic expression of the ectodomain of TMPRSS2, as shown by glucosylation of rac and by cell rounding. Pull

down expts showed a specificity of TMPSRR2 for TcsH but not the related toxin TcsL, suggesting that TcsL does not use TMPSRR2 as a receptor. The phylogenetic analysis of TMRPSS2 is un-surprising and need not be shown in this figure. The specificity for mouse MPRSS2 and not the other types is somewhat irrelevant and not surprising, especially as the multiple sequence alignment is not shown.

Fig 4. Fig 4a does not make any sense.

It shows that the CROP-deletion toxin has the same activity against WT and TMRPSS2 $-/-$ cells. No conclusion can be made about the role of the CROP repeats, except that in this experiment the CROPS are not required for entry into either WT or $-/-$ cells. Also Fig 4a is not referenced in the text.

However the BLI assays do show interaction in vitro between toxin containing the the CROP domains and TMRPSS2, and no interaction without the CROP domains. In all the experiments in Figure 4, the authors should show SDS-PAGE of the toxin proteins and the TMRPSSR deletion proteins to demonstrate that adequate expression and purity of these proteins. The conclusion that “TMRPSS2 may recognize TcsH via multiple regions” is not helpful and should be deleted.

No effort was made to investigate the regions of the CROPs required for recognition of either fucosylated structures or TMRPSS2. Short domains of the CROPS could be expressed in *E. coli* and used in BLI assays to probe further the regions necessary for binding. This would be a useful addition to the manuscript and would not be a considerable amount of work.

The knock-out experiments show that neither fucosylation or the presence of TMRPSS2 is required for toxin activity, because activity can be detected in their absence. However the levels if toxin required are very high (10²-10³ pM), several orders of magnitude higher than when both receptors are present. Can the authors rule out that the receptors are acting in synergy rather than acting independently?

Fig 5. It is unsurprising that, upon in vivo injection, WT toxin is more active than the CROP-repeat mutant. However this is a crude mode and it would be highly referable to construct *P. sordellii* mutants containing these derivatives and to test in a more meaningful model of infection. However I realise that these experiments are outside the scope of this manuscript and are technically challenging.

In places, the use of English needs attention.

Reviewer #3 (Remarks to the Author):

In the manuscript “Paeniclostridium sordellii hemorrhagic toxin targets TMPRSS2 to induce colonic epithelial lesions” Tao and colleagues investigate host receptors that bind to the hemorrhagic toxin TcsH. Using a CRISPR/Cas 9 screen in HT29 cell line they identify four genes that conferred resistance to toxin mediated cell rounding. Three of the genes, GMDS, FUT4 and SLC35C1, are involved in the fucosylation of cell surface glycans. Selective knockdown of each of these three genes demonstrated resistance to TcsH and that the CROP domain of TcsH was necessary for binding and intoxication. The other gene identified in the screen, TMPRSS2 bind to TcsH independent of fucosylation, though still via the CROP domain region. Direct administration of TcsH in vivo to a mouse results in acute toxemia that is ameliorated in TMPRSS2 knockout mice. The authors provide clear, convincing data that they have identified two novel cell entry pathways for TcsH. This observation of a significant interest to the field. This reviewer has the following minor critiques

Minor Critiques:

1. 6. In the abstract on line 28 reads “genetic description of...” do the authors “genetic deletion of...”?
2. The authors show successful knockdown of TMPRSS2 expression Fig 2D. Can they show the same decreased protein expression of Fut4, GMDS, SLC35C1?
3. Can the authors elaborate on the specific cell subsets that TMPRSS2 is expressed? Specifically, epithelial cell subsets?
4. Some figure legend could be improved by providing more detail of what is being displayed. For example, in Figure 3h it is unclear how is GFP being expressed and what it represents. Presumably the different TMPRSS protein?
5. Page 10 line 192-193 states that TcsH and TcsH1-1832 are equally potent in WT and TMPRSS2 KO cells. There is no figure panel referencing this result. Presumably the authors are referring to Figure 4A. However, figure 4A only shows cell rounding with TcsH1-1832, not TcsH. Can the author show the data with the TcsH? Shouldn't the WT MCF-7 cell exhibit increased susceptibility to the WT TcsH?

6. Figure 4e. Can the images of cell round be quantified (like in Fig 4d) to quantitatively support the authors conclusions on page 11, line 212-214 that TMPRSS2 expression in MCF-7 GMDS/FUT4 KO cells increases susceptibility to TcsH? Can the authors explain why TMPRSS2 overexpression sensitizes these cells if MCF-7 cells are already highly expressing TMPRSS2?

7. LCT is not defined. Presumably it stands for large clostridial toxin?

8. Since the HT29 cell line is the dominant cell line used in the CRISPR screen, the author should dedicate a few sentences describing the origin and why it is a reasonable cell line to use for their screen

Response to Reviewers (NCOMMS-22-02349-T)

We very much thank the reviewers for their elaborate comments, which help to largely improve our manuscript. Followed their suggestions, we have revised our manuscript and made the point-to-point response as below.

Reviewer #1

1) The main findings of this study are convincing and of high relevance in the field of bacterial toxins. By and large, the experiments are well-performed with appropriate controls. It is also of note that the manuscript is well-written and the figures of high quality, adhering to the high standards of the journal.

However, several issues need to be addressed, before considering acceptance of this sophisticated manuscript for publication.

Response: We very much appreciate the reviewer's kind support and detailed suggestions/comments, which largely help us to improve the manuscript.

2) Line 51 and 58: Two abbreviations (LCGT, LCT) are used for the same toxin family.

Response: We have worked through the manuscript and changed all LCGT to LCT for consistency. (Line 47)

3) Line 95 and 98: It is not entirely clear to me, what is meant with "mixed population".

Response: HT-29 Cas9 cells were transduced with gRNA expressing lentiviruses and selected with puromycin. These cells are referred to as the "mixed population", as single clones are not isolated thus each cell may contain different kinds of gene modifications (deletions, insertions, or mutations).

4) Line 129: "Extended Data Fig. 3" indicated here are not explained in the main text.

Response: We thank the reviewer for the reminder. We have added an explanation for it in the main text. (Line 131-132)

5) Line 131: "...fucosylation-mediated TcsH binding does not require TMPRSS2"; is the opposite also the case?

Response: Overexpression of Tmprss2 in the MCF-7 *GMDS*^{-/-} or *FUT4*^{-/-} cells could sensitize the cells. In addition, knocking-out TMPRSS2 in the MCF-7 *GMDS*^{-/-} cells further reduces the binding and entry of TcsH. Therefore, TMPRSS2 is capable of mediating the binding/entry of TcsH independent of fucosylation. (Line 232-240, Fig. 4e-h)

6) Line 147: "...susceptibility of the tested cells to TcsH"; such data about the

susceptibility mentioned here are missing in the manuscript; what is known about fucosylated glycans in these cell lines mentioned here?

Response: We are sorry a figure citation (Fig. 1a) is missing here. Fucosylation is broadly observed in most (if not all) cell types. We also confirmed the presence of fucosylation in some of the cells used in this study (see below). (Line 172).

7) *Line 151: Unclear, why the term Tmprss2 is in small letters here, but elsewhere in capital letters.*

Response: According to the general nomenclature rule, human gene/protein symbols usually are all in capital letters while mouse gene/protein symbols have only the first letter capitalized. To reduce the unclarity, we have modified the sentence to “ectopically expressing human TMPRSS2, TMPRSS2^{S441A}, or mouse Tmprss2 ...”. (Line 173)

8) *Line 157: “into the HT-29 cells”; why did the authors test Camostat (only) in this cell line? It is a bit confusing that in most experiments MCF-7 cells are used, but in this case HT-29 cells. Also HeLa cells are used in this manuscript, without explanation from the authors why they switched to these cells.*

Response: We performed the genetic screening in HT-29 cells. Once TMPRSS2 was obtained as a candidate, we directly tested Camostat on HT-29 cells as these cells were just ready. Surely, MCF-7 can be used in this assay. As suggested, we have performed the same assay on MCF-7 cells and the data has been added to the manuscript. (Line 159-163, Fig. 3b)

HeLa cells are used for ectopic expression of TMPRSS proteins because they naturally produce little to no TMPRSS2. We have also re-organized the paragraphs in the text. (Line 168-172)

9) *Line 158: “...Camostat did not show any protection of the cells...”; positive control for the inhibitory action of Camostat is missing. It must be shown that Camostat is functional in the concentration used here. I am not convinced from the current data.*

Response: The concentration of Camostat used for inhibiting the serine protease activity of TMPRSS2 (100 μ M) was chosen based on the previous studies for viruses (Hoffmann et al. 2021). We have validated the activity of the purchased Camostat by inhibiting the serine protease activity of trypsin in the cell culture (based on a method used by Yang et al. 2022, see below). Besides, we have generated a TMPRSS2 mutant (S441A) that lacks the serine protease activity. Both TMPRSS2 and TMPRSS2^{S441A} can normally mediate the cellular entry of TcsH, which is another strong evidence demonstrating that the serine protease activity is not needed for TcsH entry. (Ref. 24,

Fig. 3a, Supplementary Fig. 5, Line 163-166, 172-175)

10) Lines 179-186: The finding that exclusively TMPRSS2 and no other family members serve as receptor for TcsH is not very surprising, since the CRISPR/Cas9 screening method is based on single-gene knockouts. If other family members would act as receptors, isolation of a TMPRSS2 KO mutant would not be possible. This should be mentioned here or in the Discussion section.

Response: We thank the reviewer for the comments. Indeed, the CRISPR/Cas9 screening method is capable of finding a receptor even with the redundancy issue, just the screen and analysis would be more difficult. For example, our previous screen revealed FZD2 as a receptor for TcdB1, and we later showed that FZD1 and FZD7 also serve as receptors for TcdB1 (Tao et al. *Nature* 2016). Similarly, SEMA6A (only) was uncovered as a receptor for TcsL in the initial CRISPR screen, while researchers later demonstrated that SEMA6B is a redundant receptor for TcsL as well (Lee et al. *Cell* 2020).

11) In addition, it is unclear to me, why the authors did not perform this reconstitution experiment with the CRISPR/Cas9-derived, defined TMPRSS2 KO cell line? Such a reconstitution experiment is of major importance in order to exclude off-target effects in CRISPR/Cas9-derived TMPRSS2 KO cells.

Response: We thank the reviewer for the suggestion. We have performed the rescue experiment on the MCF-7 *TMPRSS2*^{-/-} cells. The new data has been added to the manuscript. (Fig. 3a, Line 154-156)

*12) Line 192: "...TcsH and TcsH1-1832 are equally potent to these cells..."; TcsH is ~1000-fold more potent in MCF-7 cells (Figure 2a), isn't it? TcsH1-1832 is equally potent in MCF-7 *GDMS*^{-/-} and *TMPRSS2*^{-/-} cells. This indicates an additional CROP-independent receptor, which should be discussed.*

Response: We are very sorry for the writing mistake and the missing citation for Fig. 4a here. This sentence should be "TcsH¹⁻¹⁸³² is equally potent to the MCF-7 WT and *TMPRSS2*^{-/-} cells (Fig. 4a), indicating the CROPs domain is essential for the recognition of *TMPRSS2*." Indeed, TcsH¹⁻¹⁸³² is equally potent to the MCF-7 WT, *TMPRSS2*^{-/-}, *GDMS*^{-/-}, and *GDMS*^{-/-}/*TMPRSS2*^{-/-} cells. (Line 204-206, 235-238, Fig. 4a)

13) Line 194: What are the two TcdA constructs shown in Figure 4b (green and purple)? Not all constructs explained in the text are shown in the Figure 4b.

Response: These are two TcdA CROPs fragments. We are sorry that they are not

explained clearly. We have added their description to the text. (Line 214-215)

14) Lines 196-197: “...interaction between *TMPRSS2* and *TcsH1-1832* was not detected.”; by which method? No data are shown. The pulldown approach shown in Figure 3e could be used to confirm the claim.

Response: We thank the reviewer for the reminder. The pulldown result shown was used to support this statement. (Line 208-209, Fig. 4b)

15) Line 197: “...sequentially similar.”; please provide the percentage in similarity.

Response: The sequential similarity between CROPs of *TcsH* and CROPs of *TcdA* is ~73%. We have added this percentage number to the text. (Line 214)

16) Line 204: “...all failed to facilitate the *TcsH* entry...”; did the authors check, if the truncated proteins are able to reach the plasma membrane and if they are expressed in similar levels? Without this information, it is not possible to know, if the deleted parts in the truncated version are relevant for the interaction with *TcsH*.

Response: We thank the reviewer for the insightful comment. As suggested, we check the cellular expression and cell surface expression of these truncated proteins. While all constructs can normally be expressed, *TMPRSS2* lacking *LDLRA* failed to reach the plasma membrane. We have rewritten this part based on the new data. (Line 219-227, Supplementary Fig. 8a-c)

17) Line 242-243: “...*TcsH* also induces severe hepatic hemorrhage and necrosis in the *Tmprss2*^{-/-} mice.”; which Figure includes this data? Images of the liver are not shown like in Extended Figure 9a for wt mice.

Response: We are sorry that the images were missing. We have added the data to the manuscript. (Line 269-271, Supplementary Fig. 10b)

18) Lines 244-246: “*TMPRSS2* is mainly expressed in certain organs/tissues such as the prostate, kidney, and gastrointestinal tracts, but not the liver.”; but why is then the liver harmed by *TcsH* as shown in Extended Figure 9a? Can the authors confirm the lack of *TMPRSS2* expression in the mouse liver by immunoblotting?

Response: As suggested by the reviewer, we have stained the liver and colon from the WT and *Tmprss2*^{-/-} mice using the *TMPRSS2* antibody. The immunohistological images clearly show that the *TMPRSS2* is highly expressed in the colonic epithelium but barely expressed in the liver. In the Discussion section, we suggest that fucosylated glycans may be responsible for the damage in the liver, yet other possibilities (e.g., unknown receptor) remain. (Line 274-276, 338-342, Supplementary Fig. 10c)

19) Line 252-253: “...alleviated pathological manifestations caused by *TcsH* were observed in the *Tmprss2*^{-/-} mice.”; since the authors used commercially available *Tmprss2*^{-/-} mice, it still would be desirable to confirm lack of *Tmprss2* expression in the colonic epithelium by immunoblotting.

Response: As suggested, we have stained the colon from the WT and *Tmprss2*^{-/-} mice

using the TMPRSS2 antibody to confirm the lack of Tmprss2 in the *Tmprss2*^{-/-} mice. Besides, it also serves as a control for the immunohistological staining in the mouse liver. (Line 274-276, Supplementary Fig. 10c)

20) *Figure 2c: Can the lectins used here compete with toxin binding and inhibit TcsH intoxication?*

Response: Following the reviewer's suggestion, we performed the toxin competition assay with the fucose-specific lectin AAL. As expected, the addition of AAL can suppress the intoxication of TcsH on MCF-7 cells. (Line 140-141, Fig. 2f)

21) *Why is there still a faint band for TMPRSS2 in the last lane for the TMPRSS2-KO mutant? This indicates that the authors might use CRISPR/Cas9-generated KO mutants that are not homozygous. T7E1 assays should be performed to clarify if the mutants are homozygous with biallelic KO of TMPRSS2.*

Response: We thank the reviewer for the comment. We also noticed the faint band and we conjecture that it could be due to the cross-reactivity of the antibody in recognizing other TMPRSS proteins. The T7E1 assay does not accurately reflect the editing efficacy in edited cells (Sentmanat et al. *Sci Rep* 2018), thus we have validated the biallelic KO for *TMPRSS2*^{-/-} (clone #3) by sequencing (see below).

```
gRNA targeting sequence
AAAGCACTGTGCATCACCTTGACCCTGGGGACCTTCCTCGTGGGAGCTGCGCTGGCCGCTGG WT TMPRSS2
TTTCGTGACACGTAGTGGAACTGGGACCCCTGGAAGGAGCACCCCTGACGCGACCGGGCGACC

AAAGCACTGTGCATCACCTTGACCCTGGGGACCTTCCTCGTGGGAGCTGCGCTGGCCGCTGG (insert 1, Δ24)
AAAGCACTGTGCATCACCTTGACCCTGGGGACCTTCCTCGTGGGAGCTGCGCTGGCCGCTGG (Δ8) | TMPRSS2-/- #3
```

22) *Figure 2e: Why does TcsH not bind to cells lacking glycan fucosylation although TMPRSS2 is expressed in those cells (Figure 2d)? Binding experiments with the constructs shown here should be repeated with the double KO (DKO) mutant shown in Figure 4f.*

Response: We think that it is due to the high abundance of fucosylated glycans on the cell surface and the amount of membrane TMPRSS2 is much less compared to glycan fucosylation. For toxin binding assays, excessive toxin (high toxin concentration) is loaded to achieve maximized/saturated cell membrane binding. In this case, TMPRSS2-mediated TcsH binding is largely masked. However, when taking the image under an overexposure condition, we could observe residual TMPRSS2-mediated TcsH binding in the GMDS KO cells, and this binding is further diminished in the DKO cells. This new data has been added to the manuscript. (Line 238-240, Fig. 4h)

23) *Figure 2f: Why did the authors not intoxicate TMPRSS2 knockout cells with TcsH1-1832? This would be of interest.*

Response: The experiment was done but we forgot to cite the figure (Fig. 4a) in the text. We have also combined the TcsH¹⁻¹⁸³² intoxication data for the MCF-7 WT,

TMPRSS2 KO, GMDS KO, and DKO cells in an integrated figure. (Fig. 4a)

24) Figure 3a: This experiment should be repeated with the MCF-7 TMPRSS2-KO cells, since HeLa cells do still express TMPRSS2 and cleavage of TMPRSS2(S441A) could occur by intermolecular cleavage. In addition, 2 nM TcsH were used in HeLa cells, a very high concentration, to achieve cell rounding already after 2 hours. At such conditions and without time kinetics, it is difficult to see any difference in the sensitivity between TMPRSS2- and TMPRSS2(S441A)-expressing HeLa cells against TcsH.

Response: We thank the reviewer for the suggestion. We have repeated this experiment using MCF-7 *TMPRSS2*^{-/-} cells. Consistent with the result obtained in HeLa, ectopic expression of TMPRSS2 and TMPRSS2^{S441A} can restore the sensitivity of MCF-7 *TMPRSS2*^{-/-} cells to TcsH. (Fig. 3a)

25) Figure 3c: The negative control TcsH: TMPRSS2(106-255) is not mentioned or described in the main text.

Response: We have added the information in the text. (Line 192-193)

26) Figure 3e. It would be great to see TcsH CROPS and TcsH1-1832 in this pull-down approach, since both constructs are available.

Response: Interaction between TMPRSS2^{106-492/R255Q} and TcsH CROPS/TcsH¹⁻¹⁸³² was later measured by BLI, so the pull-down assay was not performed previously. As requested by the reviewer, we have shown the pull-down assay result. (Fig. 4b)

27) Figure 3f: The fitting curve should be shown that was used to estimate the dissociation constant.

Response: The fitting curve for estimating the dissociation constant was shown in the Supplementary Information. (Line 197-199, Supplementary Fig. 7b)

28) Figure 3f: Please use “sec” (Fig. 3f) or “s” (Fig. 4b) in the x-axis uniformly.

Response: We have changed all “s” to “sec” for consistency.

29) Figure 4a: A citation in the text is missing.

Response: We have added the figure citation for Fig. 4a in the text. (Line 205 and 238)

30) Figure 4b: Why is the binding of TcsH1833-2618 here twice as high when compared to Extended Figure 7c?

Response: In Figure 4b (now Fig. 4c), a higher protein concentration (1 μM, the previous figure legend is wrong and has been corrected) was used to confirm that TcdA CROPs fragments do not bind TMPRSS2. The highest concentration used in Supplementary Fig. 7c is 400 nM.

31) Figure 4f: It would be great to test also TcsH1-1832 and TcdA here. Especially for TcsH1-1832, it would be interesting to know, if it is equally potent in GMDS KO cells and DKO cells when compared to wildtype cells as shown for the TMPRSS2 KO cells in Figure 4a.

Response: As suggested by the reviewer, we have included the cell-rounding curves of TcsH¹⁻¹⁸³² in the WT, GMDS KO, Tmprss2 KO, and DKO in a combined figure. (Fig. 4a)

32) *Extended Figure 7b and 7c: For uniformity reasons, please include here also the amino acid numbers of the ectodomains (after Tmprss2) as provided in Figure 3f.*

Response: As suggested, we have added the amino acid numbers of these ectodomains for consistency. (Supplementary Fig. 7b-c)

Reviewer #2

1) *Introduction. There is an assumption that all readers will know what a CRISPR screen is and the rationale for the experiments conducted. This is not necessarily the case. The authors should state that the purpose of the screens is to select for mutants that are resistant to the effects of the toxin, and that some of these mutants will be deficient in expression of a surface component that recognises TcsH toxin. It would also be helpful to mention that HT29 cells are a human colorectal adenocarcinoma cell line. Last paragraph on Introduction is poorly written. It is essentially a repeat of the essential conclusions reached in the manuscript. In addition, there is no mention in the Introduction of the species used as the experimental model.*

Response: We thank the reviewer for comments on the Introduction section. As suggested, we have included a brief description of the CRISPR screen for bacterial cytotoxins. The last part of the Introduction section was rewritten. We have also mentioned that HT-29 is a human colorectal adenocarcinoma cell line and described the mouse model in the Introduction. (Line 66-85)

2) *.....The phylogenetic analysis of Tmprss2 is un-surprising and need not be shown in this figure. The specificity for mouse Tmprss2 and not the other types is somewhat irrelevant and not surprising, especially as the multiple sequence alignment is not shown.*

Response: We thank the reviewer for the comments. The phylogenetic analysis may help to understand the closeness of the Tmprss proteins. We agree with the reviewer that it may not be that important for the main figure, and we have moved it to the supplementary material. Mouse Tmprss2 was included in the phylogenetic tree because it was involved in the rescue experiments and *in vivo* study. (Supplementary Fig. 6)

3) *..... In all the experiments in Figure 4, the authors should show SDS-PAGE of the toxin proteins and the Tmprss2 deletion proteins to demonstrate that adequate expression and purity of these proteins. The conclusion that “Tmprss2 may recognize TcsH via multiple regions” is not helpful and should be deleted.*

Response: The purified toxin proteins and Fc-TMPRSS2^{106-492/R255Q} are shown by SDS-PAGE in Fig. 3e and 4b. TMPRSS2 truncates are expressed in the HeLa cells by transient transfection, their expressions are monitored by immunoblot and immunofluorescence. As suggested, we have also deleted the statement “TMPRSS2 may recognize TcsH via multiple regions” from the text. (Fig. 3e and 4b, Supplementary Fig. 8b-c)

4) *No effort was made to investigate the regions of the CROPs required for recognition of either fucosylated structures or TMRPSS2. Short domains of the CROPs could be expressed in E. coli and used in BLI assays to probe further the regions necessary for binding. This would be a useful addition to the manuscript and would not be a considerable amount of work.*

Response: We thank the reviewer for the suggestion. We have further investigated the short regions of CROPs required for the recognition of fucosylated structures. The new data have been included in Fig. 2g.

As for TMPRSS2-binding, because TcsH¹⁸³³⁻²⁶¹⁸ has already shown a reduced affinity compared to the full-length TcsH (0.13 nM versus 5.23 nM), we postulate that other regions in TcsH may also contribute to the binding of TMPRSS2. Therefore, we did not further test smaller CROPs fragments.

(Fig. 2g, Supplementary Fig. 3b, Line 142-150)

5) *..... Can the authors rule out that the receptors are acting in synergy rather than acting independently?*

Response: Since we have shown that TMPRSS2 and fucosylated glycan can independently mediate the binding/entry of TcsH, we guess that the reviewer is asking whether TcsH binds to TMPRSS2 and fucosylated glycans simultaneously or competitively. To investigate this, we used Fc-TMPRSS2^{106-492/R255Q} as a competitor and found that it also protected the *TMPRSS2*^{-/-} cells from TcsH, implying TMPRSS2 and fucosylated glycans are likely competitive receptors for TcsH. We have also modified the related part in the Discussion section. (Line 242-247, 323-325, Fig. 4i)

6) *In places, the use of English needs attention.*

Response: As suggested, we have carefully revised the manuscript for the wording.

Reviewer #3

1) *In the abstract on line 28 reads “genetic description of...” do the authors “genetic deletion of...”?*

Response: We have corrected the indicated typo. (Line 25)

2) *The authors show successful knockdown of TMPRSS2 expression Fig 2D. Can they show the same decreased protein expression of Fut4, GMDS, SLC35C1?*

Response: We thank the reviewer for the comment. We did not show these protein expressions because we could not find reliable antibodies for them. On the other hand, AAL and LTL are lectins commonly used for labeling fucosylated glycans, they had been used to validate the presence/absence of surface fucosylation in the study.

3) *Can the authors elaborate on the specific cell subsets that TMPRSS2 is expressed? Specifically, epithelial cell subsets?*

Response: In the gastrointestinal tract, TMPRSS2 is expressed in the epithelial cells which is reported by many studies. We also validated the Tmprss2 expression in the mouse colon by immunostaining. (Line 274-276, Supplementary Fig. 10c)

4) *Some figure legend could be improved by providing more detail of what is being displayed. For example, in Figure 3h it is unclear how is GFP being expressed and what it represents. Presumably the different TMPRSS protein?*

Response: As suggested by the reviewer, we have revised some figure legends by adding more details. GFP is co-expressed (not fused) with Tmprss proteins from the same plasmids. The genes that encode GFP and Tmprss proteins are divided by IRES.

5) *Page 10 line 192-193 states that TcsH and TcsH1-1832 are equally potent in WT and TMPRSS2 KO cells. There is no figure panel referencing this result. Presumably the authors are referring to Figure 4A. However, figure 4A only shows cell rounding with TcsH1-1832, not TcsH. Can the author show the data with the TcsH? Shouldn't the WT MCF-7 cell exhibit increased susceptibility to the WT TcsH?*

Response: We are very sorry for the writing mistake and the missing citation for Fig. 4a here. This sentence should be “TcsH¹⁻¹⁸³² is equally potent to the MCF-7 WT and TMPRSS2^{-/-} cells (Fig. 4a), indicating the CROPs domain is essential for the recognition of TMPRSS2.” (Line 204-206, Fig. 4a)

6) *Figure 4e. Can the images of cell round be quantified (like in Fig 4d) to quantitatively support the authors conclusions on page 11, line 212-214 that TMPRSS2 expression in MCF-7 GMDS/FUT4 KO cells increases susceptibility to TcsH?*

Response: As suggested, we have quantified the percentage of rounded cells in Fig. 4e. (Fig. 4f)

7) *Can the authors explain why TMPRSS2 overexpression sensitizes these cells if MCF-7 cells are already highly expressing TMPRSS2?*

Response: The MCF-7 cells express TMPRSS2 while the expression level can be further increased greatly by transient transfection of a mouse *Tmprss2*, as demonstrated by the immunoblot analysis (see below). With more receptors, the MCF-7 cells would become more sensitive to TcsH.

8) *LCT is not defined. Presumably it stands for large clostridial toxin?*

Response: Yes, LCT stands for “large clostridial toxin”. We have added the full term to the manuscript when it first appears. (Line 47)

9) *Since the HT29 cell line is the dominant cell line used in the CRISPR screen, the author should dedicate a few sentences describing the origin and why it is a reasonable cell line to use for their screen.*

Response: We thank the reviewer for the comment. As suggested, we have indicated that the HT-29 is a human colorectal epithelial adenocarcinoma cell line and the colon lumen is a major location where *P. sordellii* colonizes in the text. (Line 91-93, 335-337)

REVIEWERS' COMMENTS

Reviewer #1 (Remarks to the Author):

The authors have answered all my criticisms through a whole series of additional experiments and convincing arguments. In my opinion, the manuscript is now ready to be published.

Reviewer #2 (Remarks to the Author):

The authors have addressed all my comments to my satisfaction.

Reviewer #3 (Remarks to the Author):

The authors have addressed this reviewer's critiques. Thank you.

Response to Reviewers (NCOMMS-22-02349-A)

We very much appreciate the reviewers for their time and support of this work. Their comments and suggestions greatly help us to improve the manuscript.

Reviewer #1

The authors have answered all my criticisms through a whole series of additional experiments and convincing arguments. In my opinion, the manuscript is now ready to be published.

Reviewer #2

The authors have addressed all my comments to my satisfaction.

Reviewer #3

The authors have addressed this reviewer's critiques. Thank you.